# Multi-agent Architecture Search via Agentic Supernet

**Guibin Zhang** [* 1]  **Luyang Niu** [* 2]  **Junfeng Fang** [1]  **Kun Wang** [3]  **Lei Bai** [4]  **Xiang Wang** [5]

## Abstract

Large Language Model (LLM)-empowered multi-agent systems extend the cognitive boundaries of individual agents through disciplined collaboration and interaction, while constructing these systems often requires labor-intensive manual designs. Despite the availability of methods to automate the design of agentic workflows, they typically seek to identify a static, complex, one-size-fits-all system, which, however, fails to dynamically allocate inference resources based on the difficulty and domain of each query. To address this challenge, we shift away from the pursuit of a monolithic agentic system, instead optimizing the **agentic supernet**, a probabilistic and continuous distribution of agentic architectures. We introduce **MaAS**, an automated framework that samples query-dependent agentic systems from the supernet, delivering high-quality solutions and tailored resource allocation (*e.g.*, LLM calls, tool calls, token cost). Comprehensive evaluation across six benchmarks demonstrates that MaAS **(I)** requires only $6 \sim 45\%$ of the inference costs of existing handcrafted or automated multi-agent systems, **(II)** surpasses them by $0.54\% \sim 16.89\%$, and **(III)** enjoys superior cross-dataset and cross-LLM-backbone transferability. The code is available at `https://github.com/bingreeky/MaAS`.

## 1. Introduction

Large Language Model (LLM)-based agents (Richards & et al., 2023; Nakajima, 2023; Reworkd, 2023) have made remarkable strides in a spectrum of domains, such as question answering (Zhu et al., 2024a), data analysis (Hong et al., 2024; Li et al., 2024), code generation (Shinn et al., 2023), web navigation (Deng et al., 2024), and data synthesis (Butt et al., 2024), by equipping LLMs with high-level features, including persona (Wang et al., 2023b; Chen et al., 2024), tools (Shen et al., 2024; Richards & et al., 2023), planning (Qiao et al., 2024; Wu et al., 2024; He et al., 2023), and memory (Zhong et al., 2024; Hatalis et al., 2023; Packer et al., 2023). Building upon the success of single agents, researchers have demonstrated that combining multiple agents, either cooperatively (Zhuge et al., 2024) or competitively (Zhao et al., 2023), can surpass the cognitive and intellectual capabilities of individuals (Du et al., 2023; Liang et al., 2023; Wang et al., 2023b; Jiang et al., 2023; Wu et al., 2023; Zhang et al., 2024a), showcasing the collective intelligence in a society of LLM-agents (Piatti et al., 2024).

Early multi-agent systems, such as CAMEL (Li et al., 2023), AutoGen (Wu et al., 2023), and MetaGPT (Hong et al., 2023), while delivering specialized capacity, often heavily rely on manual configurations, including prompt engineering, agent profiling, and inter-agent communication pipelines (Qian et al., 2024). This dependency significantly limits the rapid adaptation of multi-agent systems to diverse domains and application scenarios (Tang et al., 2023; Zhang et al., 2024c). More recently, the research community has shifted toward automating multi-agent system design. For instance, DsPy (Khattab et al., 2023) and Evo-Prompting (Guo et al., 2023) automate prompt optimization, GPTSwarm (Zhuge et al., 2024) and G-Designer (Zhang et al., 2024b) optimize inter-agent communication, and EvoAgent (Yuan et al., 2024) and AutoAgents (Chen et al., 2023a) self-evolve agent profiling. Nevertheless, they typically focus on automating specific aspects of the system. Subsequently, ADAS (Hu et al., 2024a), AgentSqure (Shang et al., 2024), and AFlow (Zhang et al., 2024c) broaden the design search space. These state-of-the-art (SOTA) methods optimize a *single, complex (multi-)agent workflow* for a given dataset via different search paradigms, *e.g.*, heuristic search (Hu et al., 2024a), Monte Carlo tree search (Zhang et al., 2024c), and evolution (Shang et al., 2024), surpassing the performance of manually designed systems.

Although the paradigm of searching for a one-size-fits-all multi-agent system appears sufficient to optimize performance-related metrics such as *accuracy* and *pass@k*, its performance is largely constrained on resource-related

---
[*]Equal contribution [1]National University of Singapore [2]Tongji University [3]Nanyang Technological University [4]Shanghai AI Laboratory [5]University of Science and Technology of China. Correspondence to: Kun Wang <wang.kun@ntu.edu.sg>, Lei Bai <baisanshi@gmail.com>.

*Proceedings of the $42^{nd}$ International Conference on Machine Learning*, Vancouver, Canada. PMLR 267, 2025. Copyright 2025 by the author(s).

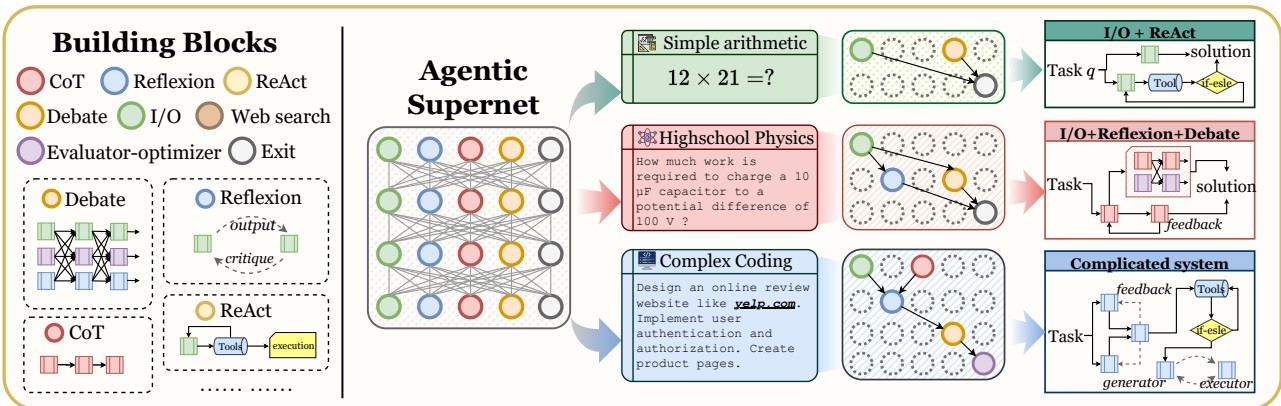

Figure 1. (**Left**) The building blocks of MaAS; (**Right**) When confronting different queries, the agentic supernet adaptively samples tailored multi-agent architecture in a query-dependent manner.

metrics, such as token cost, LLM calls, and inference latency (**Dilemma 1**). Specifically, contemporary methods tend to optimize for a complex and resource-intensive agentic system, often involving dozens of LLM API calls and external tool usage (Liu et al., 2023). However, this is far from an optimal solution: for example, in mathematical benchmarks (Hendrycks et al., 2021), Ph.D.-level abstract algebra may indeed require complicated, token-heavy systems, while simple elementary-level arithmetic works well with a single zero-shot I/O. This paradigm becomes even more problematic when applied to benchmarks across multiple task domains (**Dilemma 2**): for instance, in the GAIA benchmark (Mialon et al., 2023), there is no single system that is optimal for both *file reading* and *web searching* tasks, leaving practitioners with no alternative but to split the benchmark and optimize separately (Zhuge et al., 2024). These dilemmas unveil that, the paradigm of automatically optimizing a single multi-agent architecture fails to meet the dynamic and evolving demands of agentic deployment.

To address the above challenges, we propose **Multi-agent Architecture Search (MaAS)**, which, instead of searching for a plausible (possibly non-existent) optimal solution, generates a *distribution* of multi-agent systems. Technically, we model the optimization of MaAS on the **agentic supernet**, a probabilistic, continuous agentic architecture distribution that encompasses a vast number of possible multi-agent candidates. The agentic supernet can be seen as a cascaded multi-layer workflow, including ❶ multiple agentic operators (*e.g.*, CoT (Wei et al., 2022), Multi-agent Debate (Du et al., 2023), ReAct (Yao et al., 2023)), as well as ❷ the parameterized probability distributions of operators across layers. During training, MaAS leverages a **controller** network to sample multi-agent architectures conditioned on input queries. The distribution parameters and operators are jointly updated based on environmental feedback, with the former's gradients approximated via Monte Carlo sampling and the latter's via textual gradient estimation.

During inference, for different queries, MaAS samples a suitable multi-agent system delivering satisfactory resolution and appropriate inference resources, thereby achieving task-customized collective intelligence.

We conduct comprehensive evaluations on seven widely adopted benchmarks, covering diverse use cases in code generation (HumanEval, MBPP), mathematical reasoning (GSM8K, MATH, SVAMP), and diverse tool usage (GAIA). Empirical results demonstrate that MaAS is ❶ **high-performing**, surpassing existing handcrafted or automated multi-agent systems by $0.54\% \sim 16.89\%$; ❷ **token-economical**, outperforming the SOTA baseline AFlow on the MATH benchmark with $15\%$ of the training cost and $25\%$ of the inference cost; ❸ **transferable** across datasets and LLM-backbones; ❹ **inductive**, demonstrating strong generalizability to unseen agentic operators.

Briefly put, our key contributions are summarized as follows:

- *Paradigm Reformulation*: We introduce the concept of **agentic supernet**, a probabilistic, continuous agentic architecture distribution, which transforms the paradigm of optimizing a single optimal multi-agent system into optimizing the distribution of multiple architectures.

- *Practical Solution*: We propose MaAS, an agentic supernet-based framework that automatically evolves powerful multi-agent systems and adaptively allocates high-performing and resource-efficient solutions for user queries with varied difficulty, domain and features.

- *Experimental Evaluation*: Extensive evaluations on six benchmarks demonstrate that our framework discovers novel agentic systems with $0.54\% \sim 16.89\%$ higher performance, significantly lower training/inference costs, transferability across benchmarks and LLMs, and superior inductive capacity.

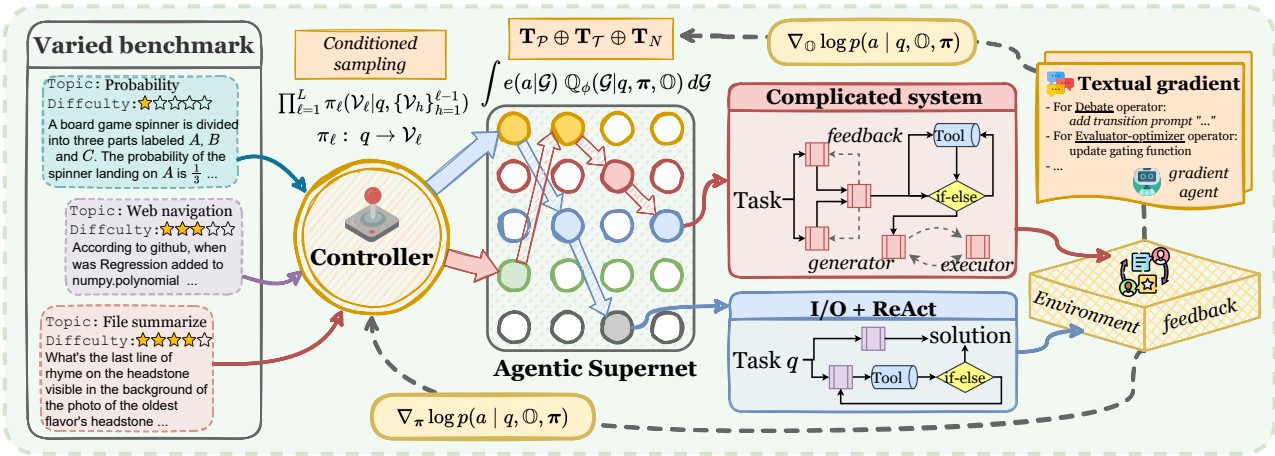

Figure 2. The overall framework of our proposed MaAS.

## 2. Related Work

**LLM-Agents and Agentic Systems.** Building on the success of single agents (Shen et al., 2024; Zhu et al., 2024b; Zhong et al., 2024), studies have shown that grouping multiple LLM-based agents into multi-agent systems (MAS) can substantially enhance individual model capabilities (Wang et al., 2024a), as demonstrated in early attempts such as AutoGen (Wu et al., 2023), LLM-Debate (Du et al., 2023), and AgentVerse (Chen et al., 2023b). However, they heavily relied on manually crafted designs, which constrained the adaptability and flexibility of agents in addressing unforeseen challenges (He et al., 2023; Chen et al., 2023b). As a result, automated agentic system design has gained increasing attention in the academic community.

**Automating Agentic Systems.** Efforts to automate the design of agent-based systems can be broadly classified into the following categories: **(I) Prompt Optimization**, such as PromptBreeder (Fernando et al., 2023), DsPy (Khattab et al., 2023), and EvoPrompt (Guo et al., 2023); **(II) Inter-agent Communication**, which focuses on orchestrating interactions between agents, including GPTSwarm (Zhuge et al., 2024), DyLAN (Liu et al., 2023), EvoMAC (Hu et al., 2024b), AgentPrune (Zhang et al., 2024a) and G-Designer (Zhang et al., 2024b); and **(III) Agent Profiling**, represented by AgentVerse (Chen et al., 2023b), EvoAgent (Yuan et al., 2024), and AutoAgents (Chen et al., 2023a). Further, ADAS (Hu et al., 2024a) and AgentSquare (Shang et al., 2024) provide more comprehensive automation for single-agent design, while AFlow (Zhang et al., 2024c) achieves multi-agent workflow automation using Monte Carlo tree search (MCTS). However, these high-performing methods still follow the paradigm of searching for a single final system, whereas MaAS searches for distribution of architectures with lower average inference costs (LLM calls, token cost, *etc.*).

**AutoML.** Automating the design of agentic systems is an emerging topic, yet the history of AutoML (He et al., 2021) provides clear precedents. Notably, the progression of agentic automation mirrors that of neural architecture search (NAS) (Ren et al., 2021). Core NAS techniques, such as reinforcement learning (Zoph, 2016), evolutionary algorithms (Liu et al., 2021), Bayesian optimization (BO) (White et al., 2021), and MCTS (Wang et al., 2021), have inspired analogous approaches in agentic automation, from policy gradient in (Zhuge et al., 2024) to evolutionary search in (Yuan et al., 2024), BO in (Shang et al., 2024), and MCTS in (Zhang et al., 2024c). In NAS, however, these black-box methods were eventually eclipsed by efficient supernet training (White et al., 2023), culminating in seminal works like DARTS (Liu et al., 2018) and SNAS (Xie et al., 2018). Inspired by this, we introduce the first MAS searching framework leveraging an ***agentic supernet***, posing new paradigms and challenges for agentic automation.

## 3. Methodology

Figure 2 illustrates the overall workflow of our method. MaAS takes diverse and varying difficulty queries as input and leverages a *controller* to sample a subnetwork from the agentic supernet for each query, corresponding to a customized multi-agent system. After the sampled system executes the query, MaAS receives environment feedback and jointly optimizes the supernet's parameterized distribution and agentic operators. In the following sections, Section 3.1 formally defines the search space and optimization objective of MaAS, Section 3.2 details how the controller query-dependently samples multi-agent structures, and Section 3.3 details the optimization of MaAS.

### 3.1. Preliminary

**Search Space.** We first define the basic unit of MaAS's search space, namely the agentic operator as follows:

**Definition 3.1** (**Agentic Operator**). An agentic operator $\mathcal{O}$ is a composite LLM-agent invocation process that involves multiple LLM calls and tool usage:

$$\mathcal{O} = \{\{\mathcal{M}_i\}_{i=1}^m, \mathcal{P}, \{\mathcal{T}_i\}_{i=1}^n\}, \\ \mathcal{M}_i \in \mathbb{M}, \mathcal{P} \in \mathbb{P}, \mathcal{T}_i \in \mathbb{T}, \tag{1}$$

where $\mathcal{M}$ and $\mathbb{M}$ correspond to LLM backbones and the set of available LLMs, respectively. Similarly, $\mathcal{P}$ and $\mathcal{T}$ represent prompts and tools. $m$ and $n$ denote the number of LLM-agents and tools invoked in the operator, respectively.

Most existing single/multi-agent workflows can be viewed as agentic operators: CoT (Wei et al., 2022) can be considered one with $m = 1$ and $n = 0$, denoted as $\mathcal{O}_{\text{CoT}}$; Self-RAG (Asai et al., 2023) similarly involves $m = 1$ agent allocation, but is equipped with $n = 1$ retrieval engine, denoted as $\mathcal{O}_{\text{SRAG}}$; Multi-agent debate (Du et al., 2023) involves multiple LLM-agent, multi-turn calls, denoted as $\mathcal{O}_{\text{Debate}}$. The feasible set of agentic operators is denoted as $\mathbb{O}$, and we discuss the initialization of $\mathbb{O}$ in Section 4.1 and Appendix B.1. We define a multi-agent system as:

$$\mathcal{G} = \{\mathcal{V}, \mathcal{E}\}, \ \mathcal{V} \subset \mathbb{O}, \ \mathcal{E} \in \mathcal{V} \times \mathcal{V}, \tag{2}$$

where $\mathcal{V}$ is the set of selected operators in $\mathcal{G}$ and $\mathcal{E}$ denotes their connectivity. $\mathcal{G}$ is constrained as a direct acyclic graph (DAG). Finally, we define the agentic supernet:

**Definition 3.2** (**Agentic Supernet**). The agentic supernet is denoted as $\mathcal{A} = \{\boldsymbol{\pi}, \mathbb{O}\} = \{\{\pi_\ell(\mathcal{O})\}_{\mathcal{O} \in \mathbb{O}}\}_{\ell=1}^L$, where:

$$\pi_\ell(\mathcal{O}) = p(\mathcal{O} \mid \mathcal{A}_{1:\ell-1}), \ \mathcal{O} \in \mathbb{O}, \\ \mathcal{A}_{1:\ell-1} = \{\{\pi_k(\mathcal{O})\}_{\mathcal{O} \in \mathbb{O}}\}_{k=1}^{\ell-1}, \tag{3}$$

where $\pi_\ell(\mathcal{O})$ represents the probability of operator $\mathcal{O}$ present at layer $\ell$, conditioned on the preceding layers $\mathcal{A}_{1:\ell-1}$. The supernet induces a joint distribution over all possible multi-layer operator configurations:

$$p(\mathcal{G}) = \prod_{\ell=1}^L \prod_{\mathcal{O} \in \mathbb{O}} \pi_\ell(\mathcal{O})^{\mathbb{I}_{\mathcal{O} \in \mathcal{V}_\ell}}, \tag{4}$$

where $\mathbb{I}_{\mathcal{O} \in \mathcal{V}_\ell}$ is the indicator function for the inclusion of $\mathcal{O}$ in the set of active operators $\mathcal{V}_\ell$ at layer $\ell$.

**Problem Formulation.** Given a benchmark $\mathcal{D}$ comprising multiple queries $q$ and their corresponding oracle answers/solutions $a$, the objective of MaAS is not to identify a single optimal agentic system like previous practices (Zhang et al., 2024c; Zhuge et al., 2024), but to optimize a conditional probability distribution as follows:

$$\max_{\mathbb{P}(\mathcal{G}|q)} \mathbb{E}_{\substack{(q,a)\sim\mathcal{D}, \\ \mathcal{G}\sim\mathbb{P}(\mathcal{G}|q)}} \big[U(\mathcal{G}; q, a) - \lambda \cdot C(\mathcal{G}; q)\big], \ \text{s.t.} \ \mathcal{G} \subset \mathcal{A} \tag{5}$$

where $\mathbb{P}(\mathcal{G}|q)$ is a distribution that generates query-dependent agentic architectures. $U(\cdot)$ and $C(\cdot)$ represent the utility/performance and cost of $\mathcal{G}$ for query $q$, respectively, and $\lambda$ is a trade-off parameter.

### 3.2. Agentic Architecture Sampling

The core of MaAS lies in tailoring a customized multi-agent system for each user query, which may vary in difficulty and domain, to deliver a satisfactory solution:

$$p(a|q, \boldsymbol{\pi}, \mathbb{O}) = \int e(a|\mathcal{G}) \, \mathbb{Q}_\phi(\mathcal{G}|q, \boldsymbol{\pi}, \mathbb{O}) \, d\mathcal{G}, \tag{6}$$

where $\mathbb{Q}_\phi$ represents the *controller network*, which takes the query $q$, the parameterized distribution $\boldsymbol{\pi}$, and the available operators $\mathbb{O}$, and outputs the sampled agentic architecture $\mathcal{G}$. $\mathbb{Q}_\phi$ is parameterized by $\phi$, and $e(\cdot|\cdot)$ denotes producing solution via executing $\mathcal{G}$. we implement $\mathbb{Q}_\phi$ as follows:

$$\mathbb{Q}_\phi(\mathcal{G}|q, \boldsymbol{\pi}, \mathbb{O}) = \prod_{\ell=1}^L \pi_\ell(\mathcal{V}_\ell|q, \{\mathcal{V}_h\}_{h=1}^{\ell-1}), \tag{7}$$

where $\mathcal{V}_h$ denotes the selected operators at layer $h$. The selection of $\mathcal{V}_\ell$ is conditionally dependent on the query $q$ and the operators from the previous layers. However, not all queries require execution across $L$ layers. As discussed in Section 1, many questions can be resolved with a simple zero-shot I/O (Zhang et al., 2024b), rendering $L$ layers unnecessarily redundant. To address this, we introduce an **early-exit operator**, denoted as $\mathcal{O}_{\text{exit}}$. During sampling, if $\mathcal{O}_{\text{exit}}$ is encountered, the process exits early:

$$\mathbb{Q}_\phi(\mathcal{G}|q, \boldsymbol{\pi}, \mathbb{O}) = \prod_{\ell=1}^L \Big[\pi_\ell(\mathcal{V}_\ell|q, \{\mathcal{V}_h\}_{h=1}^{\ell-1}) \cdot \mathbb{I}_{\mathcal{O}_{\text{exit}} \notin \mathcal{V}_\ell}\Big] \\ + \mathbb{I}_{\mathcal{O}_{\text{exit}} \in \mathcal{V}_\ell} \cdot \delta\Big(\ell - \ell_{\text{exit}}\Big), \tag{8}$$

where $\ell_{\text{exit}}$ denotes the layer at which $\mathcal{O}_{\text{exit}}$ appears, and $\delta(\cdot)$ is the Kronecker delta function. We implement the sampling process $\pi_\phi$ with a Mixture-of-Expert (MoE)-style network (Shazeer et al., 2017; Huang et al., 2024):

$$\pi_\ell: \ q \to \mathcal{V}_\ell, \ \mathcal{V}_\ell = \{\mathcal{O}_{\ell 1}, \mathcal{O}_{\ell 2}, \cdots, \mathcal{O}_{\ell t}\}, \\ t = \underset{k \in \{1, \cdots, |\mathbb{O}|\}}{\arg\min} \sum_{j < k} \mathbf{S}_k^\downarrow > thres, \tag{9}$$

where $\mathbf{S}^\downarrow = \text{sort}(\mathbf{S}, \text{desc})$, and $\mathbf{S} \in \mathbb{R}^{|\mathbb{O}|} = [S_1, \cdots, S_{|\mathbb{O}|}]$ represents the activation scores of all feasible operators w.r.t. $q$. Note that $thres$ is a threshold value that governs operator activation. Operators are activated sequentially, starting from the one with the highest score, and the process continues until the cumulative score exceeds $thres$. This ensures that the number of selected operators per layer is query-dependent, allowing MaAS to dynamically allocate resources based on task complexity. $\mathbf{S}$ is given by: $S_i = \text{FFN}(\mathbf{v}(q) \| \sum_{\mathcal{O} \in \mathcal{V}_1} \mathbf{v}(\mathcal{O}) \| \cdots \| \sum_{\mathcal{O} \in \mathcal{V}_{\ell-1}} \mathbf{v}(\mathcal{O}))$, where $\mathbf{v}(\cdot)$ denotes the embedding function using lightweight models like MiniLM (Wang et al., 2020) and Sentence-Bert (Reimers, 2019), and $\|$ represents concatenation. The detailed implementation of $\mathbf{v}(\cdot)$ is placed in Appendix B.2.

Upon completing the sequential sampling procedure in MaAS, a task-specific multi-agent system $\mathcal{G}$ is generated and executed to produce the answer $\tilde{a}$. In the next section, we elucidate the process of updating the agentic supernet based on environmental feedback.

### 3.3. Cost-constrained Supernet Optimization

We present the optimization objective of MaAS as follows:

$$\min_{\boldsymbol{\pi},\mathbb{O}} \mathbb{E}_{(q,a)\sim\mathcal{D},\mathcal{G}\sim\mathbb{Q}_\phi} \left[ -p(a|q,\boldsymbol{\pi},\mathbb{O}) + \lambda \cdot C(\mathcal{G};q) \right] \quad (10)$$

where $C(\cdot)$ evaluates the cost of multi-agent systems, represented by token cost, and $\lambda$ is the trade-off parameter. The term $p(a|q,\boldsymbol{\pi},\mathbb{O})$ in Equation (10) corresponds to Equation (6), where the calculation of $e(a|\mathcal{G})$ often involves external tools or API-based LLM calls, rendering it non-differentiable. Therefore, we employ an empirical Bayes Monte Carlo procedure (Carlin & Louis, 2000; Yan et al., 2021) to estimate the gradient w.r.t the distribution $\boldsymbol{\pi}$:

$$\nabla_{\boldsymbol{\pi}}\mathcal{L} \approx \frac{1}{K} \sum_{(q,a)\in\mathcal{D}} \sum_{k=1}^{K} \left[ m_k \nabla_{\boldsymbol{\pi}} p(\mathcal{G}_k) \right],$$
$$m_k = \frac{p(a|q,\mathcal{G}_k)}{\sum_i p(a|q,\mathcal{G}_i)} - \lambda \cdot \frac{C(\mathcal{G}_k;q)}{\sum_i C(\mathcal{G}_i;q)}, \quad (11)$$

where $m_k$ denotes the cost-aware importance weights of the agentic architecture. Intuitively, the distribution $\boldsymbol{\pi}$ is updated to favor multi-agent systems that generate high-quality solutions with minimal token cost.

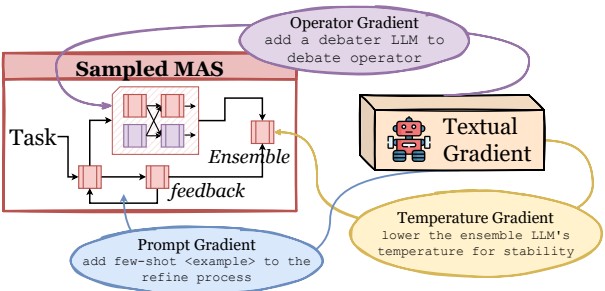

*Figure 3.* The demonstration of textual gradient.

However, the gradient w.r.t operators $\nabla_{\mathbb{O}}\mathcal{L}$ cannot be computed similarly. As shown in Equation (1), operators include black-box tool usage and natural language prompts, making numerical gradient updates infeasible. To address this, we utilize agent-based textual gradient (Hao et al., 2023; Liu et al., 2023; Hu et al., 2024b; Zhou et al., 2024) to approximate the backpropagation for agentic operators, as visualized in Figure 3 and formalized as follows:

$$\nabla_{\mathbb{O}}\mathcal{L} = \mathbf{T}_\mathcal{P} \oplus \mathbf{T}_\mathcal{T} \oplus \mathbf{T}_N, \mathbf{T}_x \in \mathbb{T}, x \in \{\mathcal{P}, \mathcal{T}, N\} \quad (12)$$

where $\mathbf{T}_\mathcal{P}, \mathbf{T}_\mathcal{T}, \mathbf{T}_N$ represent agent-generated gradient analyses in textual format, corresponding to updates of the prompt, model temperature, and operator node structure

---

**Algorithm 1** Algorithm workflow of MaAS

**Input** : A dataset $\mathcal{D}$ containing training set $\mathcal{D}_{\text{train}}$ and test set $\mathcal{D}_{\text{test}}$, Operator set $\mathbb{O}$, Randomly initialized distribution $\boldsymbol{\pi}$, Controller network $\mathbb{Q}_\phi$

**Output** : Well-optimized agentic supernet, composed of distribution $\boldsymbol{\pi}$ and operators $\mathbb{O}$

**for** $(q, a)$ in $\mathcal{D}_{\text{train}}$ **do**
  /* Sample query-dependent MAS      */
  **for** layer $\ell \leftarrow 1$ **to** $L$ **do**
    $\mathcal{V}_\ell \leftarrow \pi_\phi(\mathcal{V}_\ell | q, \{\mathcal{V}_h\}_{h=1}^{\ell-1});$      ▷ Eq. 9
    **if** $\ell = L$ or $\mathcal{O}_{\text{exit}} \in \mathcal{V}_\ell$ **then**
      break// Exit when reaching maximal sampling depth or encountering the early-exit operator

  Obtain $\mathcal{G} \leftarrow \langle \mathcal{V}_1, \cdots, \mathcal{V}_\ell \rangle$ for query $q$;      ▷ Eq. 8
  /* Execute sampled MAS      */
  Execute $\mathcal{G}$ and obtain $\tilde{a} \leftarrow e(a|\mathcal{G})$;      ▷ Eq. 6
  /* Self-evolve agentic supernet   */
  Compute loss w.r.t. $\boldsymbol{\pi}$, $\nabla_{\boldsymbol{\pi}}\mathcal{L}$;      ▷ Eq. 11
  Estimate loss w.r.t $\mathbb{O}$ via textual gradient;  ▷ Eq. 12
  Update $\boldsymbol{\pi}$ and $\mathbb{O}$ accordingly;      ▷ Eq. 10

---

(such as merging, splitting, altering, *etc.*), respectively. See prompts in Appendix B.3. In this way, the core components of the agentic supernet, namely the agentic operators and their connectivity, are jointly updated, enabling the fully automated evolution of multi-agent systems. We summarize the notations in Table 5, and the algorithm in Algorithm 1.

## 4. Experiments

### 4.1. Experiment Setup

**Tasks and Benchmarks.** We evaluate MaAS on six public benchmarks covering three domains: **(1) math reasoning**, GSM8K (Cobbe et al., 2021), MATH (Hendrycks et al., 2021), and MultiArith (Roy & Roth, 2016); **(2) code generation**, HumanEval (Chen et al., 2021) and MBPP (Austin et al., 2021)); and **(3) tool use**, GAIA (Mialon et al., 2023). For the MATH benchmark, we follow (Hong et al., 2024) in selecting 617 problems from four typical problem types (Combinatorics & Probability, Number Theory, Pre-algebra, Pre-calculus). The dataset statistics are in Appendix C.1.

**Baselines.** We compare MaAS with three series of agentic baselines: **(1) single agent execution methods**, including CoT (Wei et al., 2022), ComplexCoT (Fu et al., 2022), Self-Consistency (Wang et al., 2023a); **(2) hand-craft multi-agent systems**, including MultiPersona (Wang et al., 2023b), LLM-Debate (Du et al., 2023), LLM-Blender (Jiang et al., 2023), DyLAN (Liu et al., 2023), AgentVerse (Chen et al., 2023b) and MacNet (Qian et al., 2024); **(3) (partially or fully) autonomous multi-agent systems**, including GPTSwarm (Zhuge et al., 2024), AutoAgents (Chen et al.,

*Table 1.* Performance comparison with single agent, hand-craft multi-agent systems, and automated agentic workflows. The base LLM is consistently set as `gpt-4o-mini` for all baselines. We **bold** the best results and underline the runner-ups.

| Method | GSM8K | MATH | MultiArith | HumanEval | MBPP | Avg. |
|---|---|---|---|---|---|---|
| Vanilla | 87.45 | 46.29 | 96.85 | 87.08 | 71.83 | 77.50 |
| CoT (Wei et al., 2022) | $87.10_{\downarrow 0.35}$ | $46.40_{\uparrow 0.11}$ | $96.31_{\downarrow 0.54}$ | $88.13_{\uparrow 1.05}$ | $71.83_{\downarrow 0.00}$ | 77.95 |
| ComplexCoT (Fu et al., 2022) | $86.89_{\downarrow 0.56}$ | $46.53_{\uparrow 0.24}$ | $96.70_{\downarrow 0.15}$ | $87.49_{\uparrow 0.41}$ | $72.36_{\uparrow 0.53}$ | 78.00 |
| SC (CoT×5) (Wang et al., 2023a) | $87.57_{\uparrow 0.12}$ | $47.91_{\uparrow 1.62}$ | $96.58_{\downarrow 0.27}$ | $88.60_{\uparrow 1.52}$ | $73.60_{\uparrow 1.77}$ | 78.85 |
| MultiPersona (Wang et al., 2023b) | $87.50_{\uparrow 0.05}$ | $45.43_{\downarrow 0.86}$ | $97.49_{\uparrow 0.64}$ | $88.32_{\uparrow 1.24}$ | $73.19_{\uparrow 1.36}$ | 78.39 |
| LLM-Debate (Du et al., 2023) | $89.47_{\uparrow 2.02}$ | $48.54_{\uparrow 2.25}$ | $97.33_{\uparrow 0.48}$ | $88.68_{\uparrow 1.60}$ | $70.29_{\downarrow 1.54}$ | 78.86 |
| LLM-Blender (Jiang et al., 2023) | $88.35_{\uparrow 0.90}$ | $46.92_{\uparrow 0.63}$ | $97.29_{\uparrow 0.44}$ | $88.80_{\uparrow 1.72}$ | $77.05_{\uparrow 5.22}$ | 79.68 |
| DyLAN (Liu et al., 2023) | $89.98_{\uparrow 2.53}$ | $48.63_{\uparrow 2.34}$ | $97.12_{\uparrow 0.27}$ | $90.42_{\uparrow 3.34}$ | $77.30_{\uparrow 5.47}$ | 80.69 |
| AgentVerse (Chen et al., 2023b) | $89.91_{\uparrow 2.46}$ | $47.35_{\uparrow 1.06}$ | $97.50_{\uparrow 0.65}$ | $89.29_{\uparrow 2.21}$ | $74.28_{\uparrow 2.45}$ | 79.67 |
| MacNet (Qian et al., 2024) | $87.95_{\uparrow 0.50}$ | $45.18_{\downarrow 1.11}$ | $96.03_{\downarrow 0.82}$ | $84.57_{\downarrow 2.51}$ | $65.28_{\downarrow 6.55}$ | 75.00 |
| AutoAgents (Chen et al., 2023a) | $87.69_{\uparrow 0.24}$ | $45.32_{\downarrow 0.97}$ | $96.42_{\downarrow 0.43}$ | $87.64_{\uparrow 0.56}$ | $71.95_{\uparrow 0.12}$ | 77.80 |
| GPTSwarm (Zhuge et al., 2024) | $89.14_{\uparrow 1.69}$ | $47.88_{\uparrow 1.59}$ | $96.79_{\downarrow 0.06}$ | $89.32_{\uparrow 2.24}$ | $77.43_{\uparrow 5.60}$ | 80.11 |
| ADAS (Hu et al., 2024a) | $86.12_{\downarrow 1.33}$ | $43.18_{\downarrow 3.11}$ | $96.02_{\downarrow 0.83}$ | $84.19_{\downarrow 2.89}$ | $68.13_{\downarrow 3.70}$ | 75.13 |
| AgentSquare (Shang et al., 2024) | $87.62_{\uparrow 0.17}$ | $48.51_{\uparrow 2.22}$ | $\underline{97.77}_{\uparrow 0.92}$ | $89.08_{\uparrow 2.00}$ | $78.46_{\uparrow 6.63}$ | 80.29 |
| AFlow (Zhang et al., 2024c) | $\underline{91.16}_{\uparrow 3.71}$ | $\underline{51.28}_{\uparrow 4.91}$ | $96.22_{\downarrow 0.63}$ | $\underline{90.93}_{\uparrow 3.85}$ | $\underline{81.67}_{\uparrow 9.84}$ | $\underline{82.25}$ |
| **MaAS (Ours)** | $\mathbf{92.30}_{\uparrow 4.85}$ | $\mathbf{51.82}_{\uparrow 5.53}$ | $\mathbf{98.80}_{\uparrow 1.95}$ | $\mathbf{92.85}_{\uparrow 5.77}$ | $\mathbf{82.17}_{\uparrow 10.34}$ | **83.59** |

*Table 2.* Performance on GAIA benchmark. The best and runner-up results are **bolded** and underlined, respectively.

| Method | Level 1 | Level 2 | Level 3 | Avg. |
|---|---|---|---|---|
| GPT-4o-mini | 7.53 | 4.40 | 0 | 4.65 |
| GPT-4 | 9.68 | 1.89 | 2.08 | 4.05 |
| AutoGPT | 13.21 | 0 | 3.85 | 4.85 |
| TapeAgent | $\underline{23.66}$ | 14.47 | **10.20** | 16.61 |
| Sibyl | 21.51 | 15.72 | 4.08 | 15.61 |
| AutoAgents | 16.13 | 0 | 0 | 5.16 |
| GPTSwarm | 23.66 | 16.35 | 2.04 | $\underline{16.33}$ |
| ADAS | 13.98 | 4.40 | 0 | 6.69 |
| AgentSquare | 22.58 | 15.72 | 6.25 | $\underline{16.34}$ |
| AFlow | 10.75 | 8.81 | 4.08 | 8.00 |
| MaAS | **25.91** | **22.01** | $\underline{6.25}$ | **20.69** |

2023a), ADAS (Hu et al., 2024a), AgentSquare (Shang et al., 2024) and AFlow (Zhang et al., 2024c). More details on baseline setups are provided in Appendix C.2.

**Implementation details.** We leverage both close-source LLM (`gpt-4o-mini-0718` (OpenAI, 2024)) and open-source LLM (`Qwen-2.5-72b-instruct` (Yang et al., 2024) and `llama-3.1-70b` (Dubey et al., 2024)). All models are accessed via APIs with the temperature set to 1. We set the number of layers as $L = 4$, the cost penalty coefficient $\lambda$ as $\lambda \in \{1e-3, 5e-3, 1e-2\}$, and the sampling times $K = 4$. $thres = 0.3$ for Equation (9).

### 4.2. Performance Analysis

We compare MaAS with 14 baselines on the GSM8K, MATH, MultiArith, HumanEval, and MBPP benchmarks in Table 1, and with 10 baselines on GAIA in Table 2. The following observations can be made:

**Obs.❶ MaAS achieves optimal performance across all**

**task domains.** The multi-agent system optimized by MaAS outperforms manually designed methods by an average of $3.90 \sim 6.40\%$ and existing automated methods by $2.07 \sim 8.26\%$. Overall, as for mathematical reasoning and code generation, MaAS achieves an average best score of 83.59%, demonstrating its versatility and superiority. Table 2 shows a comparison of MaAS with automated systems and three additional baselines, including Auto-GPT (Richards & et al., 2023), TapeAgent (Bahdanau et al., 2024), and Sibyl (Wang et al., 2024b) on the GAIA benchmark. GAIA encompasses tasks from various domains such as web browsing, file reading, and multimodal understanding, making it challenging to pursue a single optimal multi-agent system for all tasks. Thus, the modest improvements of AFlow and ADAS over vanilla LLMs (only $3.35\% \uparrow$ and $2.04\% \uparrow$ on average) are understandable. In contrast, MaAS can adaptively sample customized agentic systems for different domains, achieving 18.38% and 17.61% improvements on Level 1 and 2 tasks, respectively.

### 4.3. Cost Analysis

To answer RQ2, we demonstrate that **MaAS is both training/inference cost-efficient** from the following three dimensions: (1) token cost, (2) API cost, and (3) wall-clock time, as shown in Table 3 and Figure 4. We observe:

**Obs.❷ MaAS's optimization is resource-friendly.** As shown in Figure 4 *(Training Tokens)*, among the various optimization-oriented agentic workflows, MaAS achieves the highest accuracy with the least training token consumption. While AFlow's accuracy is comparable to that of MaAS, its training cost reaches 22.50\$, which is $6.8\times$ that of MaAS (merely 3.38\$). Additionally, existing agentic

*Table 3.* Efficiency comparison between MaAS and state-of-the-art baselines on the MATH Benchmark. We shade the values of the lowest token/cost/wall-clock time and the highest performance.

| Method | Training | | | | Inference | | | | Overall |
|---|---|---|---|---|---|---|---|---|---|
| | Prompt token | Completion token | Total cost ($) | Wall-clock time (min) | Prompt token | Completion token | Total cost ($) | Wall-clock time (min) | Acc. (%) |
| LLM-Debate | - | - | - | - | $3,275,764$ | $10,459,097$ | $6.76\$$ | $92$ | $48.54$ |
| DyLAN | $22,152,407$ | $16,147,052$ | $13.01\$$ | $508$ | $6,081,483$ | $3,303,522$ | $2.89\$$ | $39$ | $48.63$ |
| MacNet | - | - | - | - | $7,522,057$ | $2,043,600$ | $2.35\$$ | $47$ | $45.18$ |
| GPTSwarm | $21,325,266$ | $6,369,884$ | $7.02\$$ | $129$ | $3,105,571$ | $788,273$ | $0.93\$$ | $30$ | $47.88$ |
| AFlow | $33,831,239$ | $29,051,840$ | $22.50\$$ | $184$ | $2,505,944$ | $2,151,931$ | $1.66\$$ | $23$ | $51.28$ |
| MaAS | $3,052,159$ | $2,380,505$ | $3.38\$$ | $53$ | $1,311,669$ | $853,116$ | $0.42\$$ | $19$ | $51.82$ |

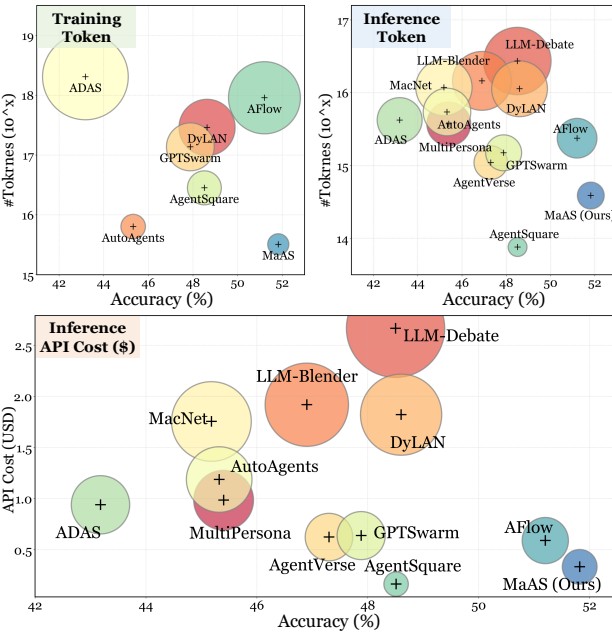

*Figure 4.* The cost analysis of MaAS on MATH benchmark.

automation pipelines are relatively time-consuming, with DyLAN taking 508 minutes and GPTSwarm taking 129 minutes. In contrast, the optimization wall-clock time of MaAS requires only 53 minutes.

**Obs.❸ Agentic supernet enjoys superior token economy during inference.** As shown in Figure 4 *(Inference API Cost)*, MaAS achieves the highest accuracy with an API cost of 0.42$, demonstrating its high performance and token economy. Although AgentSquare's API cost is slightly lower than MaAS's, this is due to its limitation to a single-agent search, which severely restricts its performance (resulting in a 4% drop compared to MaAS). Table 3 further highlights that MaAS has the lowest prompt/completion token consumption, the lowest API cost, and the shortest wall-clock time during inference. These advantages can be attributed to the agentic supernet's ability to dynamically allocate resources based on the difficulty of the query.

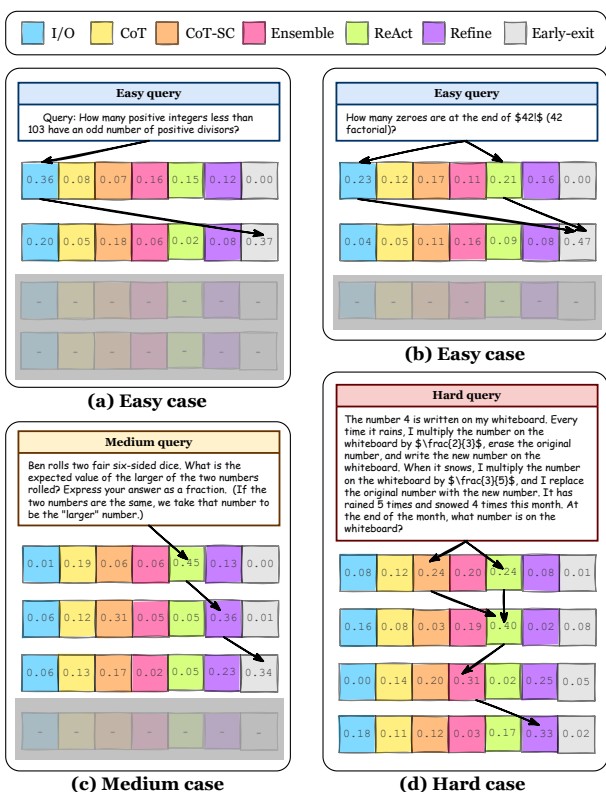

*Figure 5.* The visualization of MaAS's operator sampling process.

## 4.4. Case Study

In this section, we explore and visualize the intrinsic mechanisms of the agentic supernet. Figure 5 showcases the probability distributions of the agentic supernet when faced with different queries, and Figure 6 presents the multi-agent systems designed by MaAS for queries from the MATH, GAIA, and HumanEval benchmarks. We have:

**Obs.❹ MaAS learns to query-aware early exit from the reasoning process.** As shown in Figure 5, when faced with the easy queries (a) and (b), MaAS exits multi-agent architecture sampling at the second layer with probabilities of 0.37 and 0.47, respectively, selecting the early-exit operator. Notably, query (b) chose two agentic operators at the first

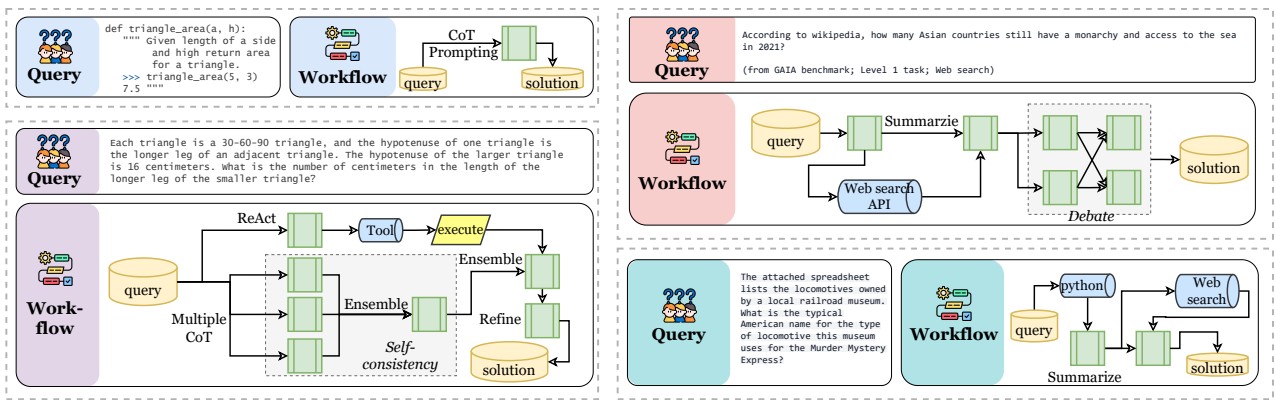

*Figure 6.* Case study and visualization for MaAS. Queries are from HumanEval, MATH and GAIA benchmarks.

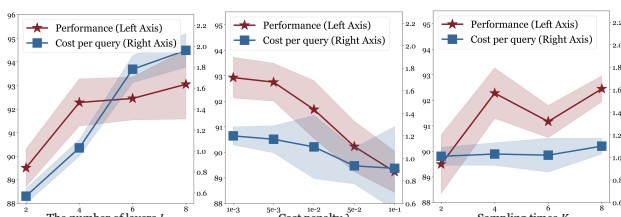

*Figure 7.* Parameter sensitivity analysis of MaAS. The unit of cost per query (right) and performance (left) is $10^{-3} \cdot \$$ and $pass@1$ (%), respectively.

layer: direct I/O and ReAct, demonstrating MaAS's ability to dynamically allocate different operators at each layer (corresponding to Equation (9)). For the more challenging queries (c) and (d), MaAS sampled additional layers, further proving its ability to customize the multi-agent system based on query awareness. This is also visualized by Figure 8, in which the probability of $\mathcal{O}_{\text{exit}}$ becomes increasingly high with the supernet depth increases.

### 4.5. Framework Analysis

**Sensitivity Analysis**   We analyze the sensitivity of MaAS to three core parameters: the number of layers in the agentic supernet $L$, the cost penalty coefficient $\lambda$ in Equation (10), and the sampling count $K$ in Equation (11). The results are presented in Figure 7. **For the parameter** $L$, we observe a significant performance improvement as $L$ increases from 2 to 4 (89.5% → 92.8%). However, further increases yield only marginal performance gains while incurring higher per-query inference costs. Considering both performance and cost, we select $L = 4$. **For the parameter** $\lambda$, we find that larger values lead MaAS to favor more cost-efficient solutions, albeit with some performance degradation. **For the parameter** $K$, we note that performance is suboptimal with highest variance when $K = 2$. Increasing $K$ to 4 effectively achieves a satisfactory low-variance estimation.

**Ablation Study**   We perform an ablation study on three key components of MaAS: **(1)** *w/o* $\nabla_{\mathbb{O}}\mathcal{L}$, removing the textual gradient in Equation (12); **(2)** *w/o* $\mathbb{O}_{\text{exit}}$, removing the

early-exit operator in Equation (8); and **(3)** *w/o* $C(\cdot)$, eliminating the cost constraint in Equation (10). We observe from Table 4 that removing the textual gradient causes the largest performance drop, as it disables MaAS's self-evolving capability. Removing $\mathbb{O}_{\text{exit}}$ and $C(\cdot)$ results in little impact on performance, but it weakens MaAS's query-dependent nature and unnecessarily increases the inference cost.

*Table 4.* Ablation study of MaAS.

| Dataset | HumanEval | | MATH | |
|---|---|---|---|---|
| Metric | Pass@1 (%) | Cost ($10^{-3}$ \$) | Accuracy (%) | Cost ($10^{-3}$ \$) |
| Vanilla MaAS | 92.85 | 1.01 | 51.82 | 0.86 |
| MaAS *w/o* $\nabla_{\mathbb{O}}\mathcal{L}$ | 90.17 | 0.90 | 48.23 | 0.84 |
| MaAS *w/o* $\mathbb{O}_{\text{exit}}$ | 91.44 | 1.67 | 51.53 | 1.04 |
| MaAS *w/o* $C(\cdot)$ | 92.94 | 1.38 | 51.19 | 1.28 |

**Transferability Analysis.**   We evaluate whether the agentic supernet of MaAS is **(1) model-agnostic** and **(2) generalizable across datasets**, with results presented in Tables 7 and 8. As shown, the agentic supernet optimized by MaAS transfers well to models such as Qwen-2.5-70b, with 4.98% ~ 5.50% ↑ in performance, while also demonstrating strong cross-dataset generalization.

**Inductive Analysis.**   To evaluate whether MaAS possesses inductive capabilities, *i.e.*, the ability to generalize to unseen agentic operators, we select the Debate (Du et al., 2023) operator as a holdout. We then compare the operator distribution of MaAS during inference with and without Debate in Figures 8 and 9. The results demonstrate that MaAS can still reasonably activate and utilize the unseen operator at an appropriate proportion.

## 5. Conclusion

In this paper, we *for the first time* shift the paradigm of automated multi-agent system design from seeking a (possibly non-existent) single optimal system to optimizing a probabilistic, continuous distribution of agentic architectures,

termed the **agentic supernet**. Building on this concept, we propose MaAS, which dynamically samples multi-agent systems that deliver satisfactory performance and token efficiency for user queries across different domains and varying levels of difficulty. We believe that MaAS paves the way toward fully automated, self-organizing, and self-evolving collective intelligence.

## Acknowledgements

This research is supported by the National Natural Science Foundation of China (No. 92270114), and the Shanghai Municipal Science and Technology Major Project.

## Impact Statement

**Ethical Considerations.** We believe that our proposed MaAS framework raises no ethical concerns regarding its motivation, design, implementation, or data usage. The method is designed to advance the automation of multi-agent systems in a principled and resource-efficient manner, ensuring responsible development while adhering to ethical guidelines in AI research.

**Societal Implications.** MaAS introduces a new paradigm in multi-agent system design by replacing static, one-size-fits-all architectures with a dynamic and adaptive agentic supernet. This approach enables fine-grained resource allocation tailored to query difficulty and domain, significantly improving efficiency while maintaining high-quality outputs. By reducing inference costs and enhancing the flexibility of multi-agent workflows, MaAS has the potential to democratize access to intelligent automation across diverse applications, including education, research, and industry.

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

# A. Notations

Table 5. Notations and Definitions

| Notation | Definition |
|---|---|
| $\mathcal{O} = \{\{\mathcal{M}_i\}_{i=1}^m, \mathcal{P}, \{\mathcal{T}_i\}_{i=1}^n\}$ | An agentic operator comprising a set of LLM instances, a textual prompt, and a set of temperature settings. |
| $\mathcal{M}$ | An individual LLM instance. |
| $\mathbb{M}$ | The set of all feasible LLMs. |
| $\mathcal{P}$ | A textual prompt used as input to the LLM. |
| $\mathbb{P}$ | The feasible space of prompts. |
| $\mathcal{T}$ | The temperature setting of the LLM. |
| $\mathbb{O}$ | The set of all feasible agentic operators. |
| $\mathcal{G} = \{\mathcal{V}, \mathcal{E}\}$ | A multi-agent system represented as a graph with vertices $\mathcal{V}$ and edges $\mathcal{E}$. |
| $\mathcal{A} = \{\boldsymbol{\pi}, \mathbb{O}\} = \{\pi_\ell(\mathcal{O})\}_{\mathcal{O}\in\mathbb{O}}^L{}_{\ell=1}$ | An $L$-layer probabilistic agentic supernet, consisting of a distribution $\boldsymbol{\pi}$ and a set of feasible operators $\mathbb{O}$. |
| $\boldsymbol{\pi}$ | The distribution associated with the agentic supernet. |
| $U(\mathcal{G}; q, a)$ | The utility evaluator of $\mathcal{G}$ with respect to query $q$ and answer $a$. |
| $C(\mathcal{G}; q, a)$ | The cost evaluator of $\mathcal{G}$ with respect to query $q$ and answer $a$. |
| $\mathbb{Q}_\phi$ | The controller network parameterized by $\phi$. |
| $e(a\|\mathcal{G})$ | Execution of $\mathcal{G}$ to produce the answer $a$. |
| $\mathcal{V}_\ell$ | The selected operators at layer $\ell$ of the agentic supernet $\mathcal{A}$. |
| $\mathcal{O}_{\text{exit}}$ | The early-exit operator. |
| $\mathbf{v}(\cdot)$ | The text embedding function. |
| $\nabla_\pi \mathcal{L}$ | The gradient of the loss $\mathcal{L}$ with respect to the distribution $\boldsymbol{\pi}$. |
| $\nabla_\mathbb{O} \mathcal{L}$ | The textual gradient of the loss $\mathcal{L}$ with respect to the operators $\mathbb{O}$. |

# B. Technical Details

### B.1. Operator Space

In this section, we detail the initialization of operator nodes as follows:

1. **Chain-of-Thought (CoT).** CoT (Wei et al., 2022) reasoning encourages the LLM to think step by step rather than directly outputting an answer. This approach enhances its capability to solve complex problems through intermediate reasoning steps, improving task handling and providing greater transparency in the decision-making process.

2. **LLM-Debate.** LLM-Debate (Du et al., 2023) allows multiple LLMs to debate, leveraging diverse perspectives to identify better solutions. In practice, we initialize three debaters and permit up to two debate rounds.

3. **Self-Consistency.** Adopting the methodology from Wang et al. (2023a), this operator aggregates five CoT reasoning paths and determines the final answer through majority voting.

4. **Self-Refine.** Following Madaan et al. (2023), this operator initially generates an answer using CoT reasoning, then prompts the agent to self-reflect iteratively. We set a maximum of five refinement iterations.

5. **Ensemble.** Inspired by LLM-Blender (Jiang et al., 2023), this operator involves three LLM-powered agents from different sources outputting answers to the same query. The pairwise ranking is used to evaluate and aggregate their responses into a final solution.

6. **Testing.** Following the test designer in AgentCoder (Huang et al., 2023), this operator is used for generating test cases for the generated code.

7. **ReAct.** Following (Yao et al., 2023), this operator enables the agent to leverage versatile tools, including code interpreter, web searching, external knowledge database, *etc.*, to handle diverse user demands.

8. **Early exit.** We introduce the early exit operator, which interrupts the multi-agent architecture sampling process and enables the depth of the agentic supernet to be query-dependent.

We respectfully note that the selection of these operators is highly customizable, allowing users the flexibility to incorporate their desired operators into the operator repository of MaAS.

## B.2. Embedding Function

Following established practices (Feng et al., 2024), we first employ an LLM to generate a comprehensive profile description for each operator. Subsequently, a lightweight text embedding model (in our case, MiniLM (Wang et al., 2020)) is used to encode the profile into a fixed-dimensional embedding. The prompt for generating the operator profile is as follows:

---
**Embedding Prompt**

```
prompt = """You are a highly proficient expert in designing and defining operators
    for large language models (LLMs). Your primary objective is to meticulously
    generate the `description` and `interface` fields for a specified operator based
    on its provided Python implementation. The generated content must be accurate,
    efficient, and precisely reflect the functionality of the operator's code.

To ensure consistency, quality, and adherence to best practices, refer to the
    following examples of previously defined operators:
{
    "Generate": {
        "description": "Generates anything based on customized input and instruction
            .",
        "interface": "generate(input: str, instruction: str) -> dict with key '
            response' of type str"
    },
    "ScEnsemble": {
        "description": "Uses self-consistency to select the solution that appears
            most frequently in the solution list, improving the selection to enhance
            the choice of the best solution.",
        "interface": "sc_ensemble(solutions: List[str], problem: str) -> dict with
            key 'response' of type str"
    }
}

Now, given the following operator code. This code encompasses the function signature
    , parameters with type annotations, internal logic, and return statements
    essential for comprehensively understanding the operator's purpose and behavior.
    Please provide its `description` and `interface` fields in the same format.
[operator code]

    """
```
---

## B.3. Textaul Gradient

The implementation of the textual gradient component is partially adapted from the repositories `https://github.com/ShengranHu/ADAS/` and `https://github.com/tsinghua-fib-lab/agentsquare`. We would like to explicitly acknowledge this contribution and express our sincere gratitude to the authors for their open-source efforts.

---
**Textual Gradient**

```
base = """
# Overview
You are an expert machine learning researcher specializing in designing agentic
    systems. Your objective is to create building blocks such as prompts and control
    flows within these systems to solve complex tasks. Specifically, you aim to
```
---

```
   design an optimal agent that performs exceptionally on the HumanEval benchmark.
   The HumanEval dataset evaluates code generation capabilities in AI systems,
   consisting of 164 hand-crafted Python programming problems. Each problem includes
   : – A function signature with a docstring describing the task – Test cases to
   verify functional correctness

# Example Question from HumanEval
[An example question from HumanEval dataset here]

# Operator code template:
class Operator:
    def __init__(self, llm: LLM, name: str):
        self.name = name
        self.llm = llm

    def __call__(self, *args, **kwargs):
        raise NotImplementedError

    async def _fill_node(self, op_class, prompt, mode=None, **extra_kwargs):
        fill_kwargs = {"context": prompt, "llm": self.llm}
        if mode:
            fill_kwargs["mode"] = mode
        fill_kwargs.update(extra_kwargs)
        node = await ActionNode.from_pydantic(op_class).fill(**fill_kwargs)
        return node.instruct_content.model_dump()

class GenerateOp(BaseModel):
    response: str = Field(default="", description="Your solution for this problem")

class Generate(Operator):
    GENERATE_PROMPT = '''
You are tasked with solving the following Python programming problem. Generate a
    complete, syntactically correct Python function that strictly adheres to the
    given requirements.

Problem:
{input}

Follow these steps:
1. Analyze the problem requirements and identify edge cases
2. Design a solution that passes all implied test cases
3. Implement the function with clear variable names and comments

Ensure:
- The code directly implements the requested functionality
- All parameters and return types match the problem specification
- Exception handling for edge cases is included when necessary '''

    def __init__(self, llm: LLM, name: str = "Generate"):
        super().__init__(llm, name)

    async def __call__(self, input: str, mode: str = None):
        prompt = self.GENERATE_PROMPT.format(input=input)
        response = await self._fill_node(GenerateOp, prompt, mode="xml_fill")
        return response

# Discovered architecture archive
Here is the archive of the discovered operator architectures:
[ARCHIVE]

# Output Instruction and Example:
The output should be a JSON object with the following structure.The first key should
    be ("thought"), and it should capture your thought process for designing the
```

```
          next operator. The second key ("description") corresponds to the brief
          description of your next operator. Finally, the last key ("code") corresponds to
          the exact operator and its prompt in Python code that you would like to try. You
          must write COMPLETE CODE in "code": Your code will be part of the entire project,
           so please implement complete, reliable, reusable code snippets.

     - thought: Captures your thought process for designing the next operator.
       - Reason about what the next interesting operator should be.
       - Describe your reasoning and the overall concept behind the operator design.
       - Detail the implementation steps.
     - description: A brief description of your next operator.
     - code: The exact operator and its prompt in Python code. Ensure the code is
          complete, reliable, and reusable.

     Here is an example of the output format for the next operator:
     [operator_example]

     You must strictly follow the exact input/output interface used above. Also, it could
          be helpful to set the LLM's role and temperature to further control the LLM's
          response. DON'T try to use some function that doesn't exist. In __call__(), you
          need to specify the instruction, input information, the prompt and the required
          output fields class for operators to do their specific part of the architecture.

     # Your task
     You are highly proficient in prompting techniques and well-versed with agentic
          systems from academic literature. Your goal is to maximize performance metrics by
           proposing innovative and effective new operators.
     Instructions:
     1. Analyze the Discovered Operators: Carefully review the operators in the archive
          to identify strengths, weaknesses, and areas for improvement.
     2. Draw Insights: Extract lessons and insights from existing operators to inform the
           design of the next operator.
     3. Innovate: Think creatively to design an operator that addresses current
          limitations or explores new functionalities, drawing inspiration from related
          agent papers or other research areas.
     4. Design the Operator: Propose the next operator's 'thought', 'description', and '
          code' following the specified format.
     5. Ensure Completeness: The generated code must be complete, reliable, and reusable,
           fitting seamlessly into the existing architecture.

     Execution Steps:
     1. Insert Operator Code: Replace the '[ARCHIVE]' and '[operator_example]'
          placeholders with actual content as needed.
     2. Generate Output: Produce the 'thought', 'description', and 'code' fields for the
          new operator, ensuring adherence to the guidelines.
     3. Validate Output: Ensure the generated JSON is correctly formatted and the code is
           syntactically and functionally correct.

     THINK OUTSIDE THE BOX and leverage interdisciplinary insights to enhance the agentic
          system's capabilities.
     """
```

## C. Experimental Details

### C.1. Dataset Statistics

Building upon established methodologies in workflow automation (Saad-Falcon et al., 2024; Hu et al., 2024a; Zhang et al., 2024c), we divide each dataset into training and test sets using a TRAIN:TEST ratio of 1:4. For the MATH benchmark, we adhere to (Hong et al., 2024), selecting a subset of 617 harder problems spanning four representative categories, Combinatorics & Probability, Number Theory, Pre-algebra, and Pre-calculus, all at difficulty level 5. The dataset statistics

are included in Table 6.

*Table 6.* Dataset Statistics.

| Domain | Dataset | #Train | #Test | Metric |
|---|---|---|---|---|
| Code Generation | HumanEval | 33 | 131 | pass@1 |
| | MBPP | 86 | 341 | pass@1 |
| Math Reasoning | GSM8K | 264 | 1055 | Accuracy |
| | MATH | 119 | 486 | Accuracy |
| | MultiArith | 150 | 600 | Accuracy |
| Tool use | GAIA | 94 | 372 | Accuracy |

### C.2. Baseline Setups

In this section, we provide a detailed description of the configurations for baseline methods:

1. **CoT.** Chain-of-Thought (CoT) prompting guides LLM agents to break down reasoning into sequential steps rather than generating direct answers. We employ the implementation from (Zhang et al., 2022).

2. **ComplexCoT.** We follow the official implementation available at `https://github.com/FranxYao/Complexity-Based-Prompting/tree/main`.

3. **Self-consistency.** To enhance robustness, we aggregate five CoT-generated solutions.

4. **LLM-Debate.** We instantiate five LLM-agents, each assigned a distinct role, which participate in up to two rounds of debate, after which the final decision is determined via majority voting. The implementation is based on `https://github.com/ucl-dark/llm_debate`.

5. **LLM-Blender.** We choose two `gpt-4o-mini`, one `Qwen-2.5-72b`, and one `llama-3.1-70b` to empower LLM-Blender (Jiang et al., 2023).

6. **DyLAN.** We directly utilize the implementation from (Liu et al., 2023).

7. **AgentVerse.** The experimental setup follows the original implementation from (Chen et al., 2023b).

8. **MacNet.** For MacNet (Qian et al., 2024), we adopt the "MacNet-MESH" variant, which corresponds to a fully connected network topology.

9. **GPTSwarm.** The method is implemented in accordance with the original settings described in (Zhuge et al., 2024).

10. **AutoAgents.** We adhere to the official configuration specified in (Chen et al., 2023a).

11. **ADAS.** The implementation details are directly inherited from (Hu et al., 2024a).

12. **AgentSquare.** We utilize the modular search framework introduced in (Shang et al., 2024). The base LLM remains fixed at `gpt-4o-mini`, with early stopping set to a patience of 5 iterations.

13. **AFlow.** In (Zhang et al., 2024c), AFlow operates with both `gpt-4o-mini` and `claude-3.5-sonnet`. To maintain fairness under homogeneous conditions, we restrict AFlow to `gpt-4o-mini` and set MAX_ITERATION=20.

## D. Supplementary Results

We have visualized the evolution of operator sampling trends as the sampling count increases, in Figure 10. MaAS learns to avoid overly confident early stopping and instead prioritizes testing and self-refinement in deeper layers.

*Table 7.* Cross-model transferability of MaAS. We optimize the agentic supernet with `gpt-4o-mini`, and report the performances before and after equipping the LLM backbones with the optimized agentic supernet.

| Dataset | HumanEval | | |
|---|---|---|---|
| LLM Backbone | `gpt-4o-mini` | `Qwen-2.5-72b` | `llama-3.1-70b` |
| vanilla | 87.08 | 85.60 | 80.06 |
| +MaAS | 92.85 | 90.14 | 85.26 |
| Dataset | MATH | | |
| LLM Backbone | `gpt-4o-mini` | `Qwen-2.5-70b` | `llama-3.1-70b` |
| vanilla | 46.29 | 63.80 | 31.93 |
| +MaAS | 51.82 | 69.35 | 42.97 |

*Table 8.* Cross-dataset transferability of MaAS. "MATH→GSM8K" denotes optimizing the agentic supernet on MATH and evaluating it on GSM8K, with similar notation applied to other cases.

| Transfer | MATH→GSM8K | GSM8K→MATH | HumanEval→ MATH |
|---|---|---|---|
| GPTSwarm | 89.96 | 45.18 | 47.92 |
| AFlow | 91.95 | 49.39 | 47.15 |
| MaAS | 92.80 | 51.02 | 50.27 |

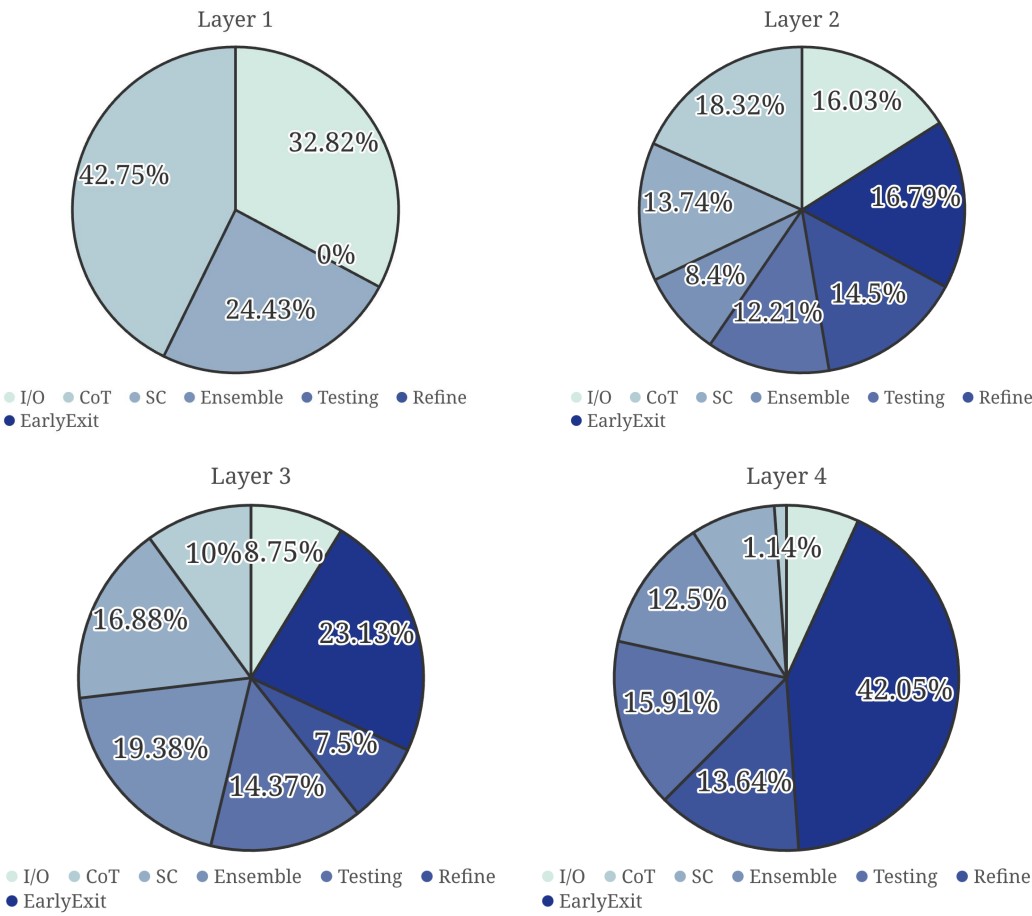

*Figure 8.* The layer-wise distribution of MaAS on HumanEval benchmark without Debate operator.

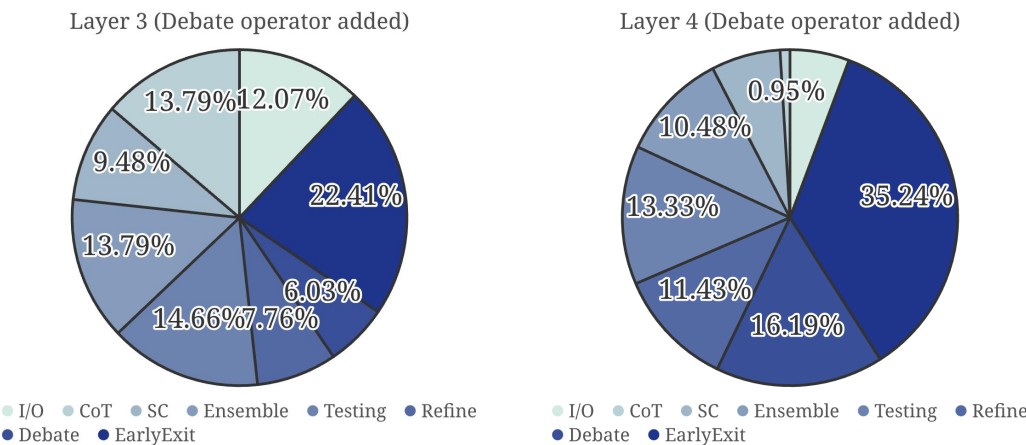

Figure 9. The layer-wise distribution of MaAS on HumanEval benchmark with Debate operator. Note that the agentic supernet is optimized with other operators, while the Debate operator is introduced only during the inference stage. It can be observed that, despite not being exposed to this operator during training, MaAS can still reasonably select it during the multi-agent architecture sampling process.

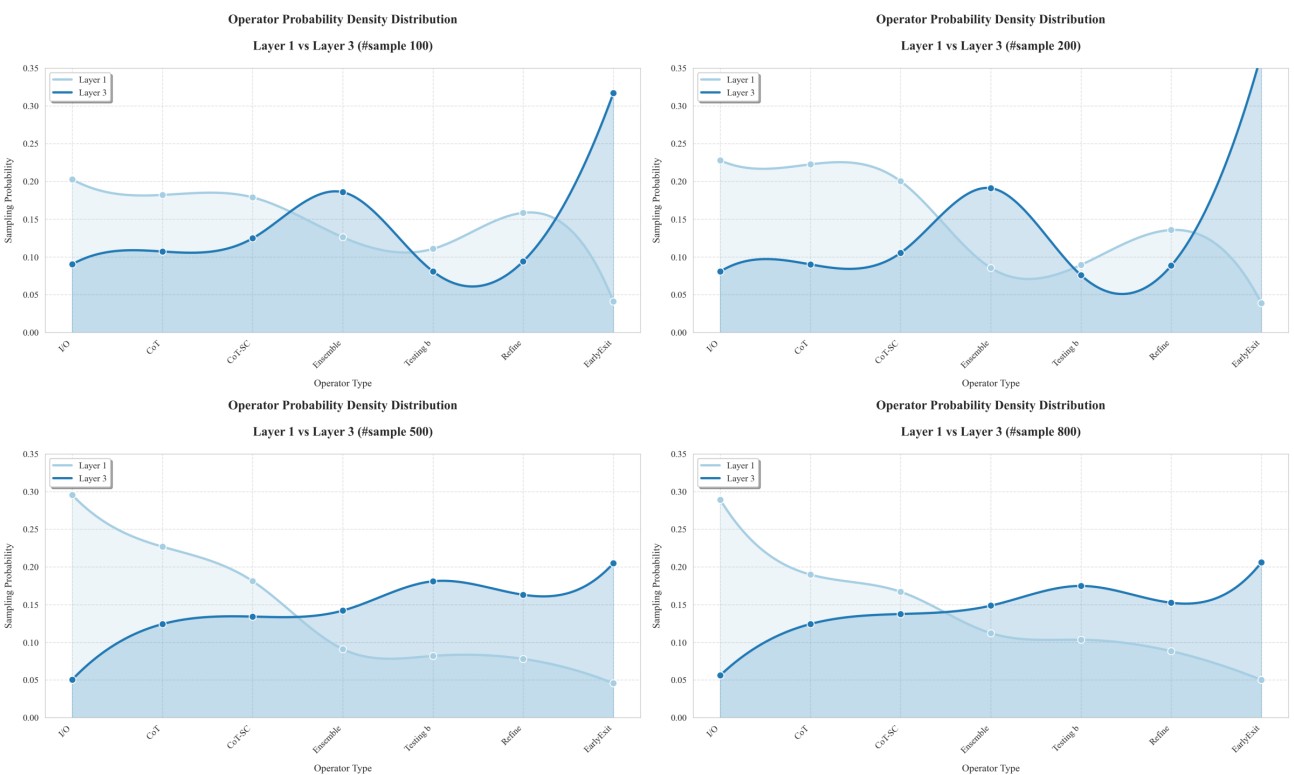

Figure 10. The evolution of operator sampling trends as the sampling count increases.

