# OpenReview forum: "Multi-agent Architecture Search via Agentic Supernet"
_ICML.cc/2025/Conference — ICML 2025 oral_

### Official Review · Reviewer_BAH6 · 2025-02-23

**Overall Recommendation:** 4

**Summary:**

This paper introduces the concept of “agentic supernet”, which transforms the automatic LLM-based multi-agents design paradigm from a static, one-size-fit-all approach to a dynamitic and adaptive framework. Their MaAS framework samples components from the supernet to assemble appropriate multi-agent systems according to the difficulty and domain of the given query. Sufficient experimental results demonstrate their method effectively reduces API costs while improving performance across a wide range of tasks and LLM backbones.

**Claims And Evidence:**

The paper is well-organized and well-written, enhancing the clarity and impact of their proposed method and findings.

**Essential References Not Discussed:**

n/a

**Experimental Designs Or Analyses:**

The authors conducted extensive experiments across multiple benchmarks to evaluate their proposed MaAS, the amount of their experiments is fair and convincing.

There appears to be a discrepancy in the reported performance improvements. The authors mentioned MaAS “surpassing existing handcrafted or automated multi-agent systems by 0.54%∼11.82%” in terms of performance, but these statistics do not in sync with the results shown in Table 1, I wonder where the value 11.82% is derived from. Besides, the improvements highlighted in red are derived from comparisons with the Vanilla baseline in Table 1, while the actual improvements over “handcrafted or automated multi-agent systems” baselines such as AFlow appear modest.

**Methods And Evaluation Criteria:**

This work presents a meaningful advancement by dynamically designing multi-agent system based on different queries through optimization of probabilistic, continuous distribution of agentic architectures, and the MaAS framework's pipeline is theoretically well-defined.

The paper lacks clarity regarding the parameter ϕ in Eq(6), where "Q_ϕ is parameterized by ϕ." The initialization and nature of this controller parameter remain unexplained.

What’s the reason the threshold value thres in Eq(9) is set as 0.3? The selection of this parameters would benefit from theoretical justification or empirical analysis, similar to the Sensitivity Analysis presented in Section 4.5.

**Other Comments Or Suggestions:**

see above

**Other Strengths And Weaknesses:**

Please refer to the weaknesses outlined above. Besides, there are some minor typos in this paper:
1. Caption of Table 1: get-4o-mini -> gpt-4o-mini
2. L81: MATh benchmark -> MATH benchmark

**Questions For Authors:**

see above

**Relation To Broader Scientific Literature:**

improve the performance on benchmark dataset

**Theoretical Claims:**

n/a

---

> ### Author Rebuttal · Authors · 2025-04-01
>
> We sincerely thank you for your careful comments and thorough understanding of our paper! Here we give point-by-point responses to your comments and describe the revisions we made to address them.
>
> ---
> >**`Weakness 1`: Clarification on parameter $\phi$** The paper lacks clarity regarding the parameter $\phi$ in Eq(6), where "$Q\_\phi$ is parameterized by $\phi$."
>
> Thank you for pointing out! The $\phi$ represents the parameters of the controller network $\mathbb{Q}\_\phi$, which is essentially composed of the sampling functions for each layer, $\pi\_l: q \rightarrow \mathcal{V}\_\ell$ (where the definition of $\pi_\ell$ is given in Eq. (9)). They are MoE-style networks that select the activated operators for the $\ell$-th layer based on the query. Therefore, $\phi$ can also be expressed as $\phi = \\{\pi\_1, \cdots, \pi\_\ell\\}$.
>
> We hope this clarifies the nature of the controller network.
>
> ---
> >**`Weakness 2: The selection of threshold value`** What’s the reason the threshold value thres in Eq(9) is set as 0.3?
>
> We supplement the parameter sensitivity analysis of threshold value in Eq. (9) as follows:
>
> |Dataset|Metric|0.1|0.2|0.3|0.4|0.5|0.6|0.7|
> |-|-|-|-|-|-|-|-|-|
> |HumanEval|Perf.|90.07|91.60|92.85|93.02|92.36|92.36|90.83|
> ||Avg. Cost|0.69|0.92|1.01|1.121|1.113|2.590|4.300|
> |GSM8K|Perf.|90.99|91.46|92.30|92.22|92.70|92.09|91.75|
> ||Avg. Cost|0.38|0.44|0.48|0.55|0.70|0.77|0.92|
>
> We observe that while increasing the threshold value leads to some performance gains, the improvement plateaus beyond 0.3. Additionally, a higher threshold increases inference costs due to the activation of more operators per layer. Therefore, we ultimately set $thres = 0.3$.
>
> ---
> >**`Weakness 3: Performance discrepancy`** There appears to be a discrepancy in the reported performance improvements.
>
> Thank you for your detailed review! After thorough inspection, we found that 11.82% was a typo, as the main text and experimental results were not properly synchronized. It should be corrected to 16.89\%, which is derived from the performance difference between MaAS and MacNet on MBPP dataset.
>
> ---
> >**`Weakness 4`** the improvements highlighted in red are derived from comparisons with the Vanilla baseline in Table 1, while the actual improvements over “handcrafted or automated multi-agent systems” baselines such as AFlow appear modest.
>
> Thank you for your insightful suggestion! We chose to present the improvements over vanilla LLMs in Table 1 following prior practices in GPTSwarm (ICML 2024) and AgentPrune (ICLR 2025), which adopt the same approach.
>
> To address your concerns, we have:
>
> 1. Provided a version of Table 1 with standard deviations, replacing the subscript values with the standard deviations from three runs, as shown in **Table1-stdev** (https://anonymous.4open.science/r/maas-rbt/table1-stdev.png).
> 2. Explained the key advantages of MaAS over the SOTA baseline AFlow:
>    - Although MaAS achieves moderate improvements over AFlow on certain benchmarks, it shows clear advantages on a broader set of benchmarks (e.g., +1.14% on GSM8K, +2.58% on MultiArith).
>    - MaAS achieves these gains with significantly lower computational costs—its training cost is only 15% of AFlow’s, and inference cost is just 25%.
>
> We sincerely hope this demonstrates MaAS’s superiority over AFlow in both cost efficiency and performance.
>
> ---
> >**`Minor Typos`**
>
> Thank you for the meticulous review! We have repaired the mentioned typos in our revised manuscript.

---

### Official Review · Reviewer_hocx · 2025-03-10

**Overall Recommendation:** 4

**Summary:**

This paper introduces MAAS (Multi-agent Architecture Search), an innovative framework for automating the design of multi-agent systems powered by Large Language Models (LLMs). MaAS addresses the limitations of existing methods that seek to identify a single, static, and complex multi-agent architecture, which often fails to dynamically allocate resources based on the difficulty and domain of each query. Instead, MaAS proposes the concept of an "agentic supernet", a probabilistic and continuous distribution of multi-agent architectures that can be sampled to tailor the system to specific queries.

**Claims And Evidence:**

The following primary claims in this paper are well-supported:

1. ​**Pursuing an agentic supernet rather than a one-size-fits-all MAS.**​ The authors advocate for optimizing their proposed agentic supernet instead of the previously attempted giant, high-latency MAS. Their experiments support this claim: MAAS outperforms the SOTA baseline AFlow with only 15% of the training cost and 25% of the inference cost. The task adaptiveness emphasized by MAAS aligns well with intuition.
2. ​**Comprehensive automation is essential for MAS.**​ The authors provide a clear and insightful overview in Section 2 of the evolution of MAS from fully manual setups to partial automation and, finally, to full automation. The mapping of the technical trajectory from neural architecture search to automated MAS is particularly interesting. The MAAS framework achieves comprehensive automation, from prompts to communication topologies.

**Essential References Not Discussed:**

I recommend that the authors include Flow[1] (ICLR 2025), which also focuses on automating agentic workflows.

[1] Flow: Modularized Agentic Workflow Automation, ICLR 2025

**Experimental Designs Or Analyses:**

MAAS is evaluated across six benchmarks, covering domains such as mathematics, coding, and tool usage. The authors also emphasize resource consumption metrics, including token count, API cost, and wall-clock time. MAAS is comprehensively demonstrated, and I find no significant flaws in the evaluation.

**Methods And Evaluation Criteria:**

The concept of the agentic supernet introduced in MAAS is novel. While it appears to be inspired by works like DARTS in NAS, it follows a completely different technical approach. The probabilistic sampling and the introduction of the early-exit operator effectively fulfill the authors’ vision of a task-dynamic MAS.

**Other Comments Or Suggestions:**

N/A

**Other Strengths And Weaknesses:**

Strength:

1. The proposed paradigm shift is innovative and significant for this field. Regardless of whether the agentic supernet becomes the mainstream form of MAS in the future, I believe this is an important contribution.
2. The evaluation is thorough, covering performance, token cost, API cost, and time consumption.

Weakness:

I find this work well-organized and convincing, with no apparent shortcomings. One thing I can suggest is to test on additional benchmarks such as ALFWorld, SciWorld, and ToolBench.

**Questions For Authors:**

See weaknesses

**Relation To Broader Scientific Literature:**

Automated MAS is an emerging and highly relevant research direction, with works like AgentVerse, GPTSwarm, and later ADAS and AFlow falling under the category of one-size-fits-all MAS. MAAS represents a new paradigm in this field and is closely connected to broader areas such as collaborative AI and autonomous AI.

**Theoretical Claims:**

N/A

---

> ### Author Rebuttal · Authors · 2025-04-01
>
> We would like to express our sincere respect for your insightful review! In response to your comments, we have carefully prepared a point-by-point reply:
>
> ---
> >**`Essential References Not Discussed`**
>
> Thank you for the valuable supplement! We have added this important citation in our revised manuscript.
>
> ---
> >**`Weaknessm 1: Additional benchmarks`** One thing I can suggest is to test on additional benchmarks such as ALFWorld, SciWorld, and ToolBench.
>
> Thank you immensely for the instructive advice! Our performance on ALFWorld is summarized in the table below. Since ALFWorld involves multiple trials, MaAS samples an architecture for each episode, meaning the same agentic workflow is used across multiple trials. Other trainable baselines, such as GPTSwarm and AFlow, are implemented similarly.
>
> | Method | Perf. (Max_trial=20) | Perf. (Max_trial=30) | Avg. Cost (10^-3 $) |
> |--------|---------------------|---------------------|-------------------|
> | Vanilla GPT-4o-mini | 48.71 | 50.12 | 4.68 |
> | CoT | 49.92 | 51.82 | 4.95 |
> | LLM-Debate | 54.68 | 56.90 | 17.30 |
> | GPTSwarm | 53.19 | 57.40 | 12.14 |
> | AgentSquare | 66.42 | 69.75 | 7.66 |
> | AFlow | 59.16 | 60.81 | 13.02 |
> | **MaAS (Ours)** | **68.14** | **72.66** | 9.15 |
>
> As observed, MaAS achieves a performance improvement of up to 22.54% on the embodied task ALFWorld while maintaining a relatively low average cost, demonstrating its cost-effectiveness.

---

> > ### Comment · Reviewer_hocx · 2025-04-03
> >
> > Thank you to the authors for the detailed response and for providing additional experiments that resolved my doubts. I have decided to increase my score and recommend accepting the paper.

---

> > > ### Author Response · Authors · 2025-04-06
> > >
> > > **Dear Reviewer `hocx`**,
> > >
> > > Thank you for your thoughtful feedback and for taking the time to review our responses. We truly appreciate your constructive insights and your willingness to engage with our clarifications. Your support of our work, especially **the recognition of our vision toward fully automated, task-dynamic MAS**, means a lot to us.
> > >
> > > Best Regards,
> > >
> > > Authors

---

### Official Review · Reviewer_43f8 · 2025-03-13

**Overall Recommendation:** 5

**Summary:**

The paper proposed a novel automated multi-agent framework through agentic supernet (MaAS), which both delivery satisfactory performance and resource allocation efficiency for user queries across different domains. The framework was comprehensively evaluated on six benchmark tasks with comparison to about 15 baselines and state-of-the-art agentic systems. Coding scripts are also provided through  anonymous GitHub.

**Claims And Evidence:**

Yes, most claims are well-supported by either reference literature or experiment results.

The only complain lies in lack of variance estimation or statistical significance of performance results in Table 1, given some of the best and runner-ups are relatively close to each other, which might weaken the claims that the proposed MaAS wins over all other baselines. The experiments are not cross-validated or randomized with multiple attempts.

**Essential References Not Discussed:**

No.

**Experimental Designs Or Analyses:**

Yes.
- It seems the experiments are not cross-validated (a single train-test split) or randomized with multiple attempts. Thus the results in table 1, 2 and 3 lack the confidence intervals. The paper is very close to strong accept if CIs are provided.
- It seems the optimal parameter set L, $\lambda$, and K are set the same for all tasks. Could you pls clarify which benchmark task yield the optimal parameter set in Figure 7? And are the optimal parameters subject to change given a different task?

**Methods And Evaluation Criteria:**

Yes. The proposed methods and evaluation criteria make sense for the problem. Evaluation metrics of benchmark datasets and tasks are well referenced. The benchmarks covers various datasets and tasks. The baseline methods include single agent system, hand-craft multi-agent systems and autonomous multi-agent systems.

Evaluation is comprehensive including not only task performances and computation costs of training and inference, but also parameter sensitivity analysis and ablation analysis.

**Other Comments Or Suggestions:**

Comments on typos and minor issues:
- Definition 3.2, missing a left parenthesis
- Missing underlines for some tied runner-ups in table 2.

Suggestions:
- It seems the cost analysis (Figure 4,Table 3), parameter sensitivity analysis (Figure 7) and ablation analysis (Table 4) is limited to MATH and HumanEval benchmarks only. If it is the case, it would be great if the authors could share the corresponding analysis for all benchmarks in supplementary for a thorough and complete benchmark analysis. If not, please help clarify in the paper if they were averaged across benchmarks.
- Some analysis in Part 4.5 Framework analysis might be too compressed even with the supplementary materials, especially transferability analysis and inductive analysis. If more details could be provided on experiment settings in supplementary, it'll be a good reference for other researchers.

**Other Strengths And Weaknesses:**

The paper is well-structured with logic flow and nice graph illustrations. I enjoy reading it very much. My questions raised during reading are well addressed in the subsequent sections. Literature review and existing state-of-art methods are well-summarized in a comparative fashion with close relation to the proposed framework. The evaluation is comprehensive with comparison to meaningful baseline methods on popular benchmark tasks. Analysis and experiment results in the supplementary also provide inspiring viewpoint in results interpretation.

**Questions For Authors:**

- How will the proposed framework scale with number of agentic operators? Could you elaborate more on the potential impact of search efficiency?
- In Figure 7, the performance of sampling times K does not monotonically increasing, e.g. dip at K=6. What's the possible reason and indication?

**Relation To Broader Scientific Literature:**

The key contributions of the paper lie in proposing a novel multi-agent framework with agentic supernet, along with comprehensive evaluations with regards to state-of-the-art agentic AI frameworks, which could potentially be a paradigm shift from seeking a single optimal system (e.g. CoT, ComplexCoT) to dynamically and autonomously optimizing a distribution of agentic architectures (various LLMs and tools). The computational efficiency of the proposed framework is also of great practical value, which could be inspiring future agentic AI research.

**Theoretical Claims:**

Not applicable.

---

> ### Author Rebuttal · Authors · 2025-04-01
>
> Sincere thanks for the thoughtful and constructive reviews of our manuscript! Based on your questions and recommendations, we give point-by-point responses to your comments.
>
> ---
> >**`Weakness 1: Lack of variance estimation`**
>
> Thank you for your insightful suggestion! In fact, all results in Table 1 represent the average of three runs. Following prior practices in GPTSwarm (ICML 2024) and AgentPrune (ICLR 2025), we chose to present the performance difference from the vanilla LLM in the main result table (rather than reporting standard deviations).
>
> To further address your concern, we provide a version of Table 1 with standard deviations, as shown in **Table1-stdev** (https://anonymous.4open.science/r/maas-rbt/table1-stdev.png). We sincerely hope this resolves your concern.
>
>
> ---
> >**`Weakness 2`** It seems the experiments are not cross-validated (a single train-test split) or randomized with multiple attempts.
>
> Thank you! The results are reported as the average of three runs, and we have supplemented them with standard deviations in our response to Weakness 1.
>
> ---
> >**`Weakness 3: Clarification on Hyperparameters`** Could you pls clarify which benchmark task yield the optimal parameter set in Figure 7? And are the optimal parameters subject to change given a different task?
>
> Thank you for your insightful inquiry! In fact, our proposed MaAS has not undergone extensive hyperparameter tuning, and the current setting ($L=4, K=4$) is not necessarily the optimal one in terms of performance. As shown in Figure 7, increasing $L$ from 4 to 8 leads to a 0.9% improvement in pass@1.
>
> However, we have consistently observed the following trends across multiple benchmarks:
>
> - Beyond $L=4$, the performance gain from increasing the supernet depth becomes marginal.
> - Beyond $K=4$, the performance gain from increasing the sampling times also plateaus.
>
> Above is the rationale behind our choice of $L=4, K=4$. Since our current sensitivity study is based on HumanEval, we further provide a sensitivity analysis on GSM8K to substantiate our findings:
>
> |$L$|2|4|6|8|
> |-|-|-|-|-|
> |Perf.|89.09|92.30|92.88|93.50|
> |Inf. Cost|0.34|0.48|0.67|0.89|
> |**K**|**2**|**4**|**6**|**8**|
> |Perf.|91.6|92.30|92.35|92.83|
> |Inf. Cost|0.48|0.48|0.49|0.49|
>
> ---
> >**`Comment 1: Typos and Minors`**
>
> Thanks immensely for pointing out! We will add the missing left parenthesis and underline the tied runner-up in Table 2.
>
> ---
> >**`Suggestion 1: Ablation & Sensitivity analysis on other benchmarks`**
>
> Thank you for your valuable insights! In our response to **Weakness 3**, we have supplemented the sensitivity analysis for GSM8K. Due to time constraints and API resource budget, we were unable to promptly provide ablation/sensitivity analyses for all datasets. However, we sincerely commit to including the corresponding analyses for other datasets in the appendix of the camera-ready version. Once again, we truly appreciate your feedback!
>
> ---
> >**`Suggestion 2: More details on transferability and inductive analysis`**
>
>
> Thank you for your valuable feedback! We sincerely commit to including additional details on transferability and inductive analysis in the appendix.
>
>
>
>
> ---
> >**`Question 1`** How will the proposed framework scale with number of agentic operators? Could you elaborate more on the potential impact of search efficiency?
>
>
> To address your question, we gradually increase the number of operators on HumanEval and report the performance, cost, and time-related metrics as follows:
>
> |operator_num|score |cost ($) |infer-time|Train-time|
> |-|-|-|-|-|
> | 3| 89.31 | 0.00093 | 6min47s | 18min |
> |4|91.6|0.00088|8min|20min|
> |5|92.13|0.00121|8min|22min|
> |6|92.36|0.00117|7min|25min|
> |7|92.85|0.00101|10min|26min|
> |8|93.89|0.00113|11min|29min|
>
>
> As observed, with more operators, performance exhibits a steady improvement, while cost remains largely stable, and training/inference time does not increase significantly. We believe this demonstrates the efficiency advantage of the agentic supernet.
>
> ---
> >**`Question 2`** In Figure 7, the performance of sampling times K does not monotonically increasing, e.g. dip at K=6. What's the possible reason and indication?
>
> Your keen insight is truly appreciated! To further investigate this phenomenon, we conducted a finer-grained hyperparameter analysis on $K$, with results summarized in the table below:
>
> |K|2|3|4|5|6|7|8|9|
> |-|-|-|-|-|-|-|-|-|
> |Perf. (3-run avg.)| 89.50 | 91.04 | 92.28 | 91.38 | 91.17 | 92.01 | 92.45 | 91.55 |
> | Perf. (5-run avg.) | 89.13 | 91.10 | 92.05 | 91.30 | 91.71 | 92.10 | 92.05 | 92.14 |
>
> The results indicate that when $K\geq4$, MaAS's performance stabilizes within the range of 91.20–92.20 on HumanEval, suggesting that the benefit of additional sampling saturates at $K=4$, with subsequent fluctuations remaining within a normal range. We hope this properly addresses your concern.

---

> > ### Comment · Reviewer_43f8 · 2025-04-04
> >
> > I appreciate the authors' detailed responses with additional experiment results for reference. The paper might be of high contribution to the agentic AI research.  I have updated my overall recommendation to strong accept.

---

> > > ### Author Response · Authors · 2025-04-06
> > >
> > > **Dear Reviewer `43f8`**,
> > >
> > > Thank you for your thoughtful feedback and strong support of our work! We greatly appreciate your constructive insights, particularly regarding variance estimation, hyperparameter configuration, and scalability. As per your suggestions, we will incorporate additional ablation studies and sensitivity analyses on more datasets in the revised manuscript. We are also sincerely grateful for your recognition of MaAS's contribution to the agentic AI community and its practical value.
> > >
> > > Thank you once again for your time, expertise, and constructive review!
> > >
> > > Best regards,
> > >
> > > Authors

---

### Official Review · Reviewer_ZEuX · 2025-03-14

**Overall Recommendation:** 5

**Summary:**

This paper introduces a novel mechanism called the "Agentic Supernet" to enable dynamic inference within multi-agent systems. Unlike traditional fixed agentic systems, the supernet and its subnet agents, which are instantiated through parameterized sampling, allowing for adaptive inference across a variety of tasks and difficulty levels. This represents a significant contribution, as it attempts to address a core limitation of workflow-based agentic systems, which often lack sufficient adaptability to diverse tasks and complexities. Extensive experimental results demonstrate the method’s effectiveness, showcasing its ability to deliver both high-performing and cost-efficient agentic systems owing to its adaptive nature.

**Claims And Evidence:**

Yes.

**Essential References Not Discussed:**

No.

**Experimental Designs Or Analyses:**

yes. All experimental designs are sound and valid.

**Methods And Evaluation Criteria:**

Yes.

**Other Comments Or Suggestions:**

- In NAS, supernets are primarily employed to reduce evaluation costs during architecture optimization. In contrast, this work leverages the supernet concept to enable dynamic inference tailored to varying queries. A detailed discussion highlighting this distinction—particularly how it shifts the focus from cost reduction to adaptability—would enrich the paper’s contribution and clarify its novelty within the broader literature.
- The choice of a feed-forward supernet is interesting. Potentially, as the information flow does not appear to strictly depend on a layer-by-layer structure, it might be also interesting to have a directed acyclic graph as a "supergraph". Including ideas like this, some future works can be discussed to inspire other researchers.
- Several baselines lack specificity regarding the underlying LLM used.
- In Figure 4, the meaning of circle size is not explained.
- Table 4 reveals an intriguing result: removing the textual gradient leads to a significantly lower-cost agentic system on HumanEval, despite a performance drop. Discussing more on this could provide valuable insights into the method’s cost-performance dynamics.
- The learning progress of the proposed algorithm is not shown. Including a visualization or analysis of the optimization process (e.g., performance convergence, operator sampling trends over time) would offer a deeper understanding of how the Agentic Supernet evolves and adapts.

**Other Strengths And Weaknesses:**

**Strengths**

- The proposal of an Agentic Supernet to facilitate adaptive inference in multi-agent systems across diverse tasks and difficulty levels is both novel and compelling. This approach introduces a fresh perspective on addressing the adaptability challenges inherent in traditional agentic frameworks.
- The experimental design is sound, following practices from prior work while expanding the scope through the inclusion of additional benchmarks and comprehensive analyses.
- The results and examples effectively demonstrate the method’s capability to adapt agent configurations to varying task difficulties, achieving high performance while maintaining cost efficiency. This underscores the practical utility of the proposed framework besides its novelty in scientific ideas.

**Weaknesses**

1. Insufficient Clarity in Presentation
   - The description of the supernet’s layer structure lacks clarity, particularly regarding whether each layer contains all operators from the operator space.
   - The process of handling multiple activated operators within a layer and how their outputs are integrated as inputs to subsequent layers remains ambiguous.

2. Drawing from supernet-related work in NAS, such as Liang et al. "DARTS+", some learned lessons can be discussed:
     - Collapse Issue: While the absence of "skip" operations in the proposed supernet may avoid the collapse issue in the proposed method, this could be validated through longer runs to confirm its robustness.
     - Overfitting is also a known challenge in NAS supernets like DARTS. Seemingly the adaptive sampling network to the query will mitigate this issue. But some discussions on it will be interesting.
     - It is also known that DARTS has a biased sampling issue (e.g. discussed in Chu et al. "Fair DARTS"). I.e. Early over-sampling of certain paths (nodes) could bias the optimization process. Because these paths (nodes) are sampled and updated at the early stage, the subsequent sampling will further favor those paths (nodes), even other paths (nodes) may have more potential. A discussion of this risk, along with potential investigation and mitigation strategies as future work, would enhance the paper’s depth.

3. The ability of the resulting agentic system to dynamically adapt to multiple tasks is a highly appealing feature, briefly evidenced in Table 8 through transferability experiments. However, further elaboration and concrete examples would provide deeper insight into this capability. Additionally, an experiment training the supernet across multiple domains simultaneously—rather than a single domain—could be interesting to reveal whether such a setup improves generalization compared to domain-specific training.

4. The paper presents two mechanisms that enable adaptive inference in agentic systems: (1) the Agentic Supernet and (2) query-based sampling during inference. The current evaluation showcases the combined effect of both (1+2), suggesting that the supernet’s effectiveness may depend on query-based sampling. However, an alternative approach—relying solely on (2) with an archive of agentic systems tailored to specific queries—could also achieve adaptive inference without the supernet. This raises a critical question: Is the supernet the primary driver of the method’s superior performance, or does the query-based sampling mechanism play the dominant role? The paper would benefit from a discussion or ablation study disentangling these contributions. Such an analysis would enhance the work’s depth and bolster the persuasiveness of the central claim regarding the supernet’s significance.

**Questions For Authors:**

See above.

**Relation To Broader Scientific Literature:**

The work is related recently emerging research topic of automating agent design. Additionally, the concept of the Agentic Supernet draws inspiration from supernet techniques in Neural Architecture Search (NAS).

**Theoretical Claims:**

No proofs.

---

> ### Author Rebuttal · Authors · 2025-04-01
>
> >**`Weakness 1: Insufficient Clarity in Presentation`**
>
> Thank you for the insightful comment! Each layer shares the same set of operators, except for the first layer, where the early-exit operator is excluded. The operators in each layer produce outputs in parallel, which are then concatenated and passed as input prompts to the next activated operator, following standard practices in Mixture-of-Agents (ICLR 2025) and GPTSwarm (ICML 2024).
>
> ---
> >**`Weakness 2: Lessons from DARTS+`**
>
> Thank you for this highly insightful discussion and enlightenment!
>
> - **Collapse Issue** We empirically demonstrate that the agentic supernet does not encounter the collapse issue observed in traditional DARTS, as presented in **Table-collapse** (https://anonymous.4open.science/r/maas-rbt/table-collapse.md).
> - **Overfitting Issue** Intuitively, we argue that MaAS's agentic supernet is inherently resistant to overfitting due to two key factors:  (1) **Query-aware sampling**. In fact, prior work on customizable NAS like GRACES [1], has demonstrated the advantage of input-dependent supernet in out-of-distribution generalizability.  (2) **Cross-domain training data**. The training data itself can span multiple domains (e.g., GAIA benchmark includes web searching and file analysis), inherently promoting cross-domain generalization.
> - **Biased Sampling Issue** We commit to incorporating this intriguing discussion in our revised manuscript, borrowing lessons from FairNAS/DARTS-/DARTS-PT.
>
> [1] Graph Neural Architecture Search Under Distribution Shifts, ICML'22
>
> ---
> >**`Weakness 3.1: Transferability of agentic supernet`**
>
> Following your suggestion, beyond the numerical transferability study in Table 8, we further visualize the underlying mechanism of MaAS’s transferability, with results and analysis presented in **Figure-transfer** (https://anonymous.4open.science/r/maas-rbt/figure-transfer.md).
>
> ---
> >**`Weakness 3.2: Cross-domain optimization of agentic supernet`**
>
> We would first like to point out that, the GAIA benchmark we used inherently falls under cross-domain optimization (web searching + file reading). To further address your concerns, we report the results of training the agentic supernet under a math/coding cross-domain setting:
>
> *(M->MATH, G->GSM8K, H->HumanEval)*
> |Train on|Test on|Perf.|
> |-|-|-|
> |M|M|51.82|
> ||G|92.80|
> |M+G|M|51.66|
> ||G|93.70|
> |H|H|92.85|
> ||M|50.27|
> |H+M|H|93.05|
> ||M|52.69|
>
>
> Notably, when trained on the HumanEval+GSM8K mixture, the performance surpasses that of training on HumanEval or MATH alone.
>
> ---
> >**`Weakness 4: Ablation on (1) supernet and (2) query-based sampling`**
>
> To validate that the driving force of MaAS relies not only on query-based sampling but also on the agentic supernet, we construct a baseline called Agent-Archive, which consists of an agentic system archive populated with operators and agentic workflows. Results are in **Table-archive** (https://anonymous.4open.science/r/maas-rbt/table-archive.md).
>
> ---
> >**`Comment 1`** How agentic supernet shifts the focus from cost reduction to adaptability
>
> We respectfully state that MaAS's adaptability stems from its query-aware supernet sampling mechanism.
>
> By analogy, MaAS removes DARTS's final one-shot pruning step that determines a fixed CNN. Instead, during actual usage, it dynamically customizes each layer’s kernel size/skip connections/pooling operators, as well as the  network depth, based on each input (e.g., image). This allows MaAS to retain DARTS's advantage of reducing training costs while simultaneously enhancing adaptability in agentic scenarios.
>
> ---
> >**`Comment 2: Possibility of supergraph`**
>
> Thank you for your inspiring thoughts! We will incorporate this interesting discussion in our updated manuscript.
>
> ---
> >**`Comment 3: Specifying underlying LLM of baselines`**
>
> Thank you! This has been specified in Line 277 and Appendix C.2. Besides, in Table 2, GPt-4o-mini is used for TapaAgent/Sibyl and GPT-4 for AutoGPT.
>
> ---
> >**`Comment 4: Circle size in Figure 4`**
>
> Thank you! The circle size in Figure 4 is proportional to the value of y-axis.
>
> ---
> >**`Comment 5: Removal of textual gradient in Table 4`**
>
>
> After carefully reviewing Table 4, we identified a typo that exaggerated the impact of removing textual gradient on inference cost. Specifically, 0.09 should be corrected to 0.90. To provide a more detailed analysis, we present Table 4 with standard deviation included for finer observation (https://anonymous.4open.science/r/maas-rbt/table4-stdev.md). We believe the cost reduction occurs because textual gradient increases the length of prompts in MaAS.
>
> ---
> >**`Comment 6: Learning visualization of agentic supernet`**
>
> Following your insightful suggestion, we have visualized the evolution of operator sampling trends as the sampling count increases (https://anonymous.4open.science/r/maas-rbt/figure-learning.pdf). It learns to avoid overly confident early stopping and instead prioritizes testing and self-refinement in deeper layers.

---

> > ### Comment · Reviewer_ZEuX · 2025-04-04
> >
> > Thank you for your thorough response to my questions. The detailed reply, along with the additional experiments, has addressed my concerns effectively. I believe this paper advances the field significantly and could inspire many exciting future research. I will raise my score and recommend acceptance.

---

> > > ### Author Response · Authors · 2025-04-06
> > >
> > > **Dear Reviewer `ZEuX`,**
> > >
> > > Thank you for your thoughtful feedback and generous support of our work! We truly appreciate your meticulous review and high-caliber suggestions, including **insights from NAS, cross-domain optimization**, and **advanced visualization**. These have significantly enriched the depth and quality of our manuscript. It has been an honor for us to incorporate your recommendations and suggestions into our revised manuscript.
> > >
> > > Sincerely,
> > >
> > > The Authors

---

### Decision · Program_Chairs · 2025-05-01

**Decision:**

Accept (oral)

**Comment:**

This paper proposes a novel paradigm—Agentic Supernet—for enabling adaptive inference in multi-agent systems. The reviewers consistently praised the originality of the framework, the comprehensive and rigorous experimental evaluation, and the practical implications for automated agent design. While some concerns were raised about clarity in architectural details and the disentangling of performance gains, the consensus is that the paper marks a significant contribution to the field. I strongly recommend acceptance.